

# Dynamics of wormhole formation in fractured karst aquifers

Wolfgang Dreybrodt[1,2] & Franci Gabrovšek[2]

[1]Faculty of Physics and Electrical Engineering, University of Bremen, Germany,
[2]Karst Research Institute ZRC SAZU, Titov trg 2, 6230 Postojna Slovenia,

*Correspondence to*: Franci Gabrovšek (gabrovsek@zrc-sazu.si)

**Abstract.** The most spectacular wormholes are caves in fractured limestone terrains. Here, a model of their evolution is presented. The modeling domain is a two-dimensional square net consisting of one-dimensional fractures with given width $b$ and length $l$. To each of the fractures one designs a constant aperture width, $a$, (homogeneous net) or an aperture width taken from a lognormal distribution (heterogeneous net). The boundary conditions are constant head $h$ at the input driving the water

downstream to the output at $h = 0$. Linear dissolution kinetics, controlled by surface kinetics and diffusion are active. First we discuss the simple case of a homogeneous net. Two steps in its evolution are observed. In the first, all fractures are widened evenly and a homogeneous even dissolution front progresses slowly into the aquifer. The second step is triggered by an instability when, due to small perturbations, some of the foremost fractures gain length compared to the neighboring ones. Then, these fractures eject flow using the parallel resistances of the net. This way they attract more fresh aggressive water and

their propagation is enhanced. Several wormholes (caves) are penetrating into the aquifer but only one reaches the output, whereas the others stop growing due to the redistribution of hydraulic heads caused by the leading wormhole. The mechanisms governing the evolution of a single wormhole are explored by increasing the aperture width of one selected input fracture by $\Delta a << a$. In this case, only one single wormhole is created and inspection of the flow rates along it reveal the mechanism of flow enhancement in detail. If one uses a heterogeneous net, the first step of evolution is suppressed because of the large

perturbations and wormholes start to grow immediately. We have also modeled the case of several competing wormholes in a homogeneous net by putting appropriate seeds. We find that there is a critical distance between the wormholes. Within this distance only one wormhole survives, whereas there is no interaction between them when they are separated by more than the critical distance. We also answer the question, Why do wormholes in a two-dimensional model exhibit breakthrough times at least one order of magnitude smaller than a one-dimensional model representing the aquifer by one single plane parallel

fracture of the same dimensions? Finally, we present several scenarios with non-homogeneous distribution of initial aperture widths. In these, uniform dissolution front does not develop and wormholes start to grow immediately, what is more likely expected in nature.





## 1 Introduction

The first numerical models of speleogenesis in terrains of soluble rock considered cave evolution only along one single isolated plane parallel fracture. First attempts using this concept failed because the linear dissolution rates used caused exponential decline of dissolution widening along the fracture. The cave conduits stopped to grow after penetrating only a few meters into the rock (Dreybrodt, 1996;White and Longyear, 1962).

The paradox that caves could not exist at all was resolved by White (1977), who suggested that a switch in the dissolution kinetics to a non-linear regime close to the equilibrium concentration of calcium ions with respect to calcite can reduce dissolution rates and causes deep penetration of dissolution power into the rock, which allows evolution of caves in geologically reasonable time. Laboratory experiments showed that this concept is valid for limestone (Svensson and Dreybrodt, 1992) and also for gypsum (Jeschke et al., 2001). Therefore, it was used in later modeling approaches of speleogenesis (Dreybrodt, 1990;Palmer, 1991), although Dreybrodt et al. (2005a) show that linear kinetics do, in fact, allow the evolution of caves if one considers simultaneously two processes: linear surface kinetics and transport of the dissolved ions by molecular diffusion into the bulk of the water.

In modeling cave genesis, homogeneous fractures with initially even spacing were used as the basic elements in two-dimensional (2D) models consisting of a net of such fractures (Dreybrodt et al., 2005a). Fracture widening depends only on the distance from the fluid inlet. Therefore, these fractures have been described as one-dimensional (1D). Using them in speleogenetic models was criticized by Szymczak and Ladd (2011). They showed that homogeneous 1D fractures exhibit an instability to infinitesimally small perturbations such that the initially evenly propagating dissolution front breaks up into channels, hereafter called "wormholes". The interaction between these wormholes causes competition, whereby only a few reach the output boundaries, while the others stop growing. This behavior is well-known from the evolution of wormholes in porous media, (Fredd and Fogler, 1998).

Szymczak and Ladd (2011) questioned the approach used commonly by many model efforts, which uses nets of 1D fractures, (e.g. Dreybrodt et al., 2005a) because the formation of wormholes within individual fractures is not taken into account. This way, the breakthrough time in an individual fracture can be reduced significantly causing a change of the hydraulic properties of the global fracture network. It is, in principle, possible to meet this criticism by discretizing each single fracture into a 2D aperture field, to permit wormhole formation, however, at high computational cost.

Alternatively, some models have used circular pipes as basic elements instead of 1D fractures (Bauer et al., 2005;Kaufmann, 2005). This avoids the formation of wormholes in the 1D elements of the 2D net. However, the results of such models are close to those using fractures. This gives confidence that the formation of wormholes in the fracture elements of the 2D net does not change the general behavior. Therefore, in this paper, we use nets of 1D fractures to investigate the formation of wormholes in 2D networks and the interaction between the evolving channels, favoring those that have gained in length compared to their neighbors. In this paper we use the term wormholes because this is common in that context. In our model caves and wormholes have the same meaning.




We will answer the following questions. Why are breakthrough times reduced even under linear kinetics when a wormhole evolves within a net of water-transmitting fractures? How does the feedback causing breakthrough in a net differ from that active in a 1D isolated fracture? How do evolving wormholes interact to select the winner and stop the competitors in further growth? How does the instability (Szymczak and Ladd (2011) influence the evolution of karst aquifers? We demonstrate that

answers can be given by applying the physical mechanisms of flow and dissolution active in a 2D net of fractures, without using complex mathematical algorithm.

## 2 The model

Here, we describe in short the model suggested by Dreybrodt et al. (2005a). First, construct a 2D square net consisting of 1D fractures with given width $b$ and length $l$. To each of the fractures assign a constant aperture width, $a$, (homogeneous net) or a

10 width taken from a lognormal distribution (heterogeneous net). Figure 1 gives an illustration (Dreybrodt et al., 2005a).

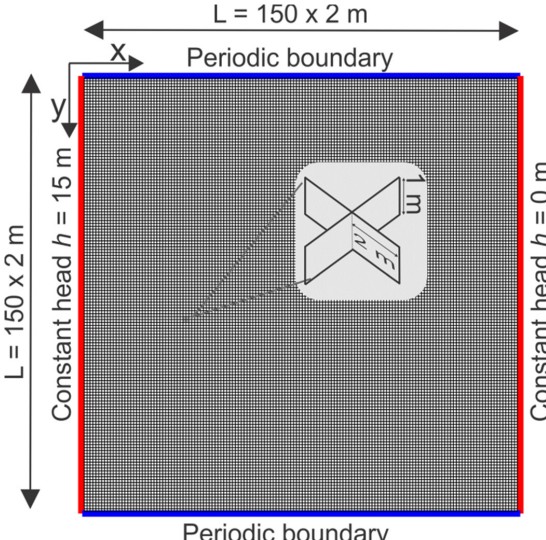

**Figure 1: Modeling domain used in this work is a rectangular grid of fractures connecting 150 x 150 nodes. Each fracture is 2 m long and 1 m wide (see the excerpt in the middle) with initial aperture width $a_0$. The left-hand and right-hand sides are constant head boundaries at 15 m and 0 m, respectively. Upper and lower boundaries are periodic. $x$-$y$ coordinate system is given for the node**
**coordinates. The logo of Copernicus Publications.**



The boundary conditions throughout this work are constant head $h = 15$ m at the input on the left-hand side ($x = 0$) and $h = 0$ m at the output on the right-hand side ($x = 150$) and symmetric boundary conditions at the upper and lower borders ($y = 0$ and $y = 150$) of the domain. The concentrations of calcium ($c$) and $CO_2$ are fixed to equal values at all input points.

In a first step, we calculate the flow in all the fractures of the net. At each confluence of fractures, we assume complete mixing of the inflowing solutions. Mass conservation requires

$$\sum Q_{in}(i) + \sum Q_{out}(i) = \sum_{j} Q_{ij} = 0 \qquad (1)$$

where $Q_{in}(i)$ is the flow rate towards node $i$ and $Q_{out}(i)$ is the flow rate away from it. $Q_{ij}$ is the flow rate through the fracture connecting nodes $i$ and $j$. For laminar flow, $Q_{ij}$ is given (Beek and Muttzall, 1975;Dreybrodt, 1996) by

$$Q_{ij} = (h_i - h_j)/R_{ij} \qquad (2)$$

$R_{ij}$ is the resistance of the fracture connecting nodes $i$ and $j$. $h_i$ and $h_j$ are the hydraulic heads at nodes $i$ and $j$. For a fracture with variable aperture width $a(x)$ along the flow direction $x$, $R_{ij}$ is given by

$$R_{ij} = \frac{12\eta}{\rho g} \int_{i}^{j} \frac{dx}{a^3(x) \cdot b} \qquad (3)$$

where $g$ is gravitational acceleration, $\rho$ is the density of water and $\eta$ is the viscosity of water. Equations 1 to 3 represent a set of linear equations for the unknown heads, which is solved by the Preconditioned Conjugate Gradients method for sparse matrices (Press et al., 2002;Stewart and Leyk, 1994).

Next, we specify the calcium concentration $c_{in} = 0$ of the inflow solution at all input points. From the $CO_2$ concentration of the inflowing water in equilibrium with a partial pressure $p_{CO2} = 0.02$ atm, the equilibrium concentration with respect to calcite, $c_{eq}$, is calculated for closed system conditions with respect to $CO_2$ to find the dissolution rates by the rate law $F(c) = k_1(1 - c/c_{eq})$ with $k_1 = 4 \cdot 10^{-11}$ mol cm$^{-2}$ s$^{-1}$ ( Dreybrodt et al., 2005) governing dissolution in the fracture draining the input points. Then, we apply the 1D transport-dissolution model (Dreybrodt, 1996), summarized shortly below, to calculate the calcium concentration profile along all fractures, including the concentration of the solution leaving it.

By following the order of decreasing heads, we select all nodes where the concentrations of all the inflowing solutions are known. We assume complete mixing of these solutions before they are transferred into conduits transporting the flow away. We repeat this until the input concentrations for all fractures are determined. From this, the new profiles of the fractures after a time step $\Delta t$ are obtained, as explained below. Then, the new flow rates are calculated and the entire procedure is repeated to obtain the temporal evolution of the net until some defined condition, such as breakthrough, is met.





### 2.1 The one-dimensional transport-dissolution model

Once the flow rate at the input of a fracture and its input concentration of calcium are known, dissolution widening of each fracture is calculated by the following procedure. The widening rate at any point in a fracture i is proportional to the dissolution rate $F(c(x))$

$$\frac{da}{dt} = 2F(c) \cdot \gamma \qquad (4)$$

where the factor $\gamma$ converts the dissolution rates from mol cm$^{-2}$ s$^{-1}$ to the retreat of the fracture wall in cm per year. Therefore, knowledge of the concentration $c(x)$ of calcium ions along the fracture is needed.

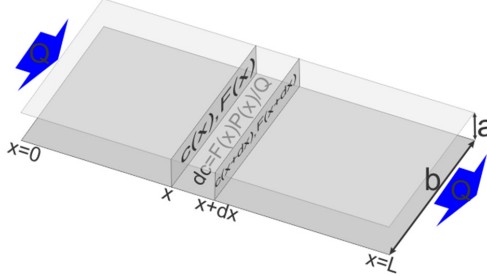

Figure 2: Discretization and mass conservation in a basic network element; a 1D fracture. The conservation of mass requires that
10 the change of concentration between x and is proportional to the surface of a section *P(x)dx*, where *P(x)* is a perimeter, to the dissolution rates *F(x)*, and inversely proportional to the flow rate Q.

Mass conservation requires that the amount of calcite dissolved from the walls during the time interval $\Delta t$ within any part of the fracture between $x$ and $x + \Delta x$ is equal to the difference between the amount of calcium leaving at $x + \Delta x$ and the amount of calcium entering at $x$ during the same time interval (see Fig. 2).

15 From this one obtains:

$$F(c(x)) \cdot P(x)\mathrm{d}x = Q\mathrm{d}c \qquad (5)$$

Integration yields

$$Q \int_{c_{in}}^{c(x)} \frac{\mathrm{d}c}{F(c)} = \int_{0}^{x} P(x)\mathrm{d}x \qquad (6)$$

For linear dissolution kinetics the dissolution rates are given by Dreybrodt et al. (2005a)

$$F(c) = k(1 - c/c_{eq}) \text{ with } k = k_1 \left(1 + \frac{k_1 a}{3Dc_{eq}}\right)^{-1} \qquad (7)$$





where $k$ is the rate constant, $c_{eq}$ is the equilibrium concentration and $k_1$ is the rate constant of the surface reaction. This takes into account the common action of surface reactions and transport by molecular diffusion. For small aperture width, $a$, $k$ is determined by surface reactions, whereas, for large $a$, $k$ is governed by diffusion (Dreybrodt, 1988). This is close to the expression of $k$ used by Szymczak and Ladd (2009).

For a uniform plane parallel fracture integration of Eq. (6) using Eq. (7) yields

$$F(x) = F(x=0) \cdot \exp(-\frac{Pk}{Qc_{eq}}x) = F(x=0) \cdot \exp(-x/\lambda), \text{ where } \lambda = (Q \cdot c_{eq})/(P \cdot k) \qquad (8)$$

$\lambda$ is the penetration length, a distance along the fracture where the concentration and the dissolution rate has dropped to $1/e = 0.37$ of its initial value at the input.

To obtain the temporal evolution of the profile of any fracture $i,j$ in the net we discretize time and spatial variables $t$ and $x$ into
suitable increments $\Delta t$ and $\Delta x$ and perform the following procedure (Dreybrodt, 1996):

1. Calculate $Q_i(t)$ by using Eqs. (1) to (3) for each fracture.

2. Calculate $F_i(x)$ from Eqs. (8) and (7).

3. Calculate the new profile assuming a constant rate in the time interval $\Delta t$ by

$$
\begin{aligned}
a(x, t+\Delta t) &= a(x,t) + 2\gamma F(c(x))\Delta t, \\
b(x, t+\Delta t) &= b(x,t) + 2\gamma F(c(x))\Delta t
\end{aligned} \qquad (9)
$$

Discretisation of the spatial variable $x$ has to be done with care. The change of concentration $\Delta c$ within the interval $\Delta t$ is given by

$$\Delta c(x,t) = \frac{F(x,t) \cdot P(x,t)}{Q_i(t)} \Delta x \qquad (10)$$

The increments $\Delta c$ are chosen such that changes in $(c_{eq} - c_i)$ are small to avoid numerical saturation (Dreybrodt, 1996). From these, suitable $\Delta x$ are obtained.



## 3. Results

### 3.1 Homogeneous net: the basic case

To get a first insight, we start with a homogeneous net with equal aperture widths $a_0 = 0.02$ cm, equal widths $b = 100$ cm, and equal lengths $l = 200$ cm for all fractures. The net, shown in Fig. 1, consists of 151 nodes in both directions, which results in a

5  dimension of 300 m by 300 m. At the left-hand input border, a hydraulic head $h = 15$ m injects flow into all its fractures. The right-hand side output border is at $h = 0$ and all fractures can drain water. The other boundaries have periodic conditions. This simulates a 1D problem, where Szymczak and Ladd (2011) found the instability with respect to infinitely small perturbations, which causes the evolution of wormholes.

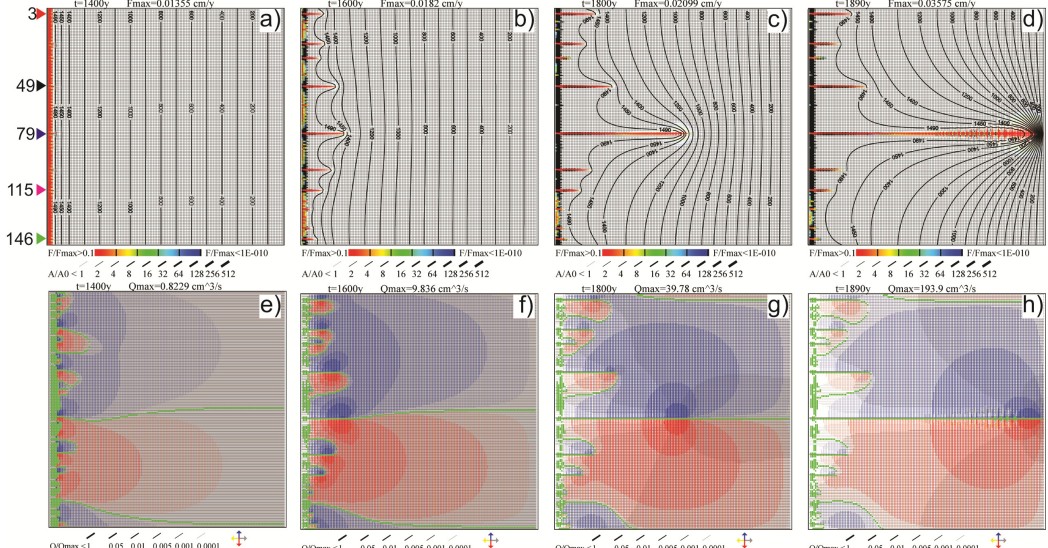

**Figure 3: Temporal evolution of a uniform fracture network. Panels in the upper row show aperture widths as line thickness and relative dissolution rates (a ratio between the rate in a fracture and the highest dissolution rate $F_{max}$ in the network) given in the color code. Lower panels show relative flow rates (ratio between the flow rate in a fracture and flow rate $Q_{max}$ in the fracture with maximal flow in the network) and flow directions as shown by the arrow code. Green lines denote the flow divides by connecting**

15  **nodes with opposite direction of outflow. White regions depict zero flow. Colored triangles mark the position of inputs where later wormholes evolve (See also Fig. 4). Numbers denote their y-coordinate.**

Figure 3 shows the temporal evolution of the aquifer. The upper panels illustrate the dissolution rates by a color code shown below and the fracture aperture widths by a bar code. The distribution of the hydraulic head is depicted by isolines in the upper panels. The lower panels depict the flow rates by the thickness of lines shown below and their directions by colors. Grey means

20  flow downstream (left–right), blue is the direction along a transverse fracture from the lower boundary to the upper one (flow up), and red is flow in the opposite direction (flow down). Note that the flow rates are normalized to the maximal rate, $Q_{max}$,


which occurs in some fracture in the net at that time. They are depicted by a bar code. White regions exhibit flow rates less than $10^{-4}Q_{max}$.

After 1400 years, an almost even front of widening channels has propagated a few meters downstream (Fig. 3a). Flow (Fig. 3e), however, exhibits an uneven distribution. There are domains of transversal flow up (blue) and down (red) originating from

the different wormholes.

After 1600 years, due to the instability caused by numerical noise, the front breaks (Fig. 3b and f) and many fractures protrude out from the previously even front. The domains of lateral flow have increased. The green color (Fig. 3e–f) connects nodes at the crests and troughs of the hydraulic potential field and, therefore, mark water divides between the lateral flows originating from the various channels.

After 1800 years (Fig. 3c and g), seven prominent channels have propagated into the net. Transversal flow from the leading channel dominates, whereas, lateral flow from the channels staying behind becomes low, as shown by the white regions (Fig. 3g). After 1890 years, only one channel has reached the output boundary, whereas all others have almost stopped growing. Due to the redistribution of the heads, the leading wormhole ejects flow perpendicular to the isolines into the net, as can be seen from the red and yellow regions of dissolution rates close to its tip in Fig. 3h. The shorter losing wormholes are now

located in a region with low hydraulic gradients. Therefore, flow in them is reduced.

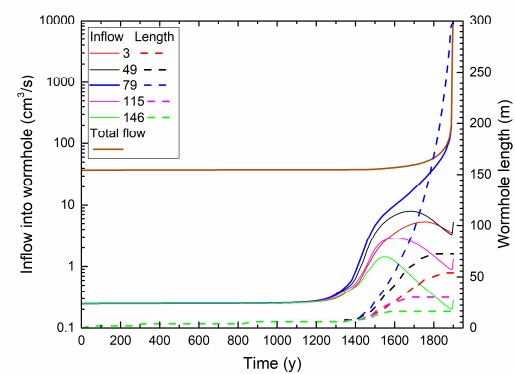

**Figure 4: Temporal evolution of the total flow through the network (solid brown line), input flow rates and lengths of wormholes at different positions (given as a y-coordinate of nodes).**

Figure 4 depicts the temporal evolution of the total flow through the domain (solid brown line), flow rates into the inputs of

the evolving wormholes (full lines) and their lengths (dashed lines) defined as the distance where $a(t)/a_0 = 2$. For the first thousand years, during the evolution of the even dissolution front, the resistance of the net changes only inside the dissolution front close to the input. All input points transmit equal flow into the net. The total flow rate (37.5 cm$^3$ s$^{-1}$) equals the flow




through one fracture (0.25 cm$^3$ s$^{-1}$) multiplied by the total number of input points (150). The flow into the input nodes of all later wormholes stays almost constant until after 1200 years when one observes a rapid increase due to the wormhole formation. Soon after, the curves separate, where more successful wormholes (at $y$ =79, 49 and 3) gain flow on behalf of the less successful ones (at $y$ = 146 and 115). The flow into latter reaches a maximum, but then drops and these wormholes stop growing. Between

1500 and 1600 years the winning wormhole ($y$ = 79), takes over the domain and increases in length until it reaches the downstream boundary of the domain. The remaining competitors are also retarded until they stop growing.

### 3.2 Homogeneous network with one seed to trigger a single wormhole

In view of the results in Fig. 3, one asks for the detailed mechanisms by which the different wormholes compete for growth and the flow they carry. To this end it would be advantageous to deal with scenarios with only a few competing wormholes.

To trigger the instability of the dissolution front, we insert seeds in the following way. Using the net of Fig. 3, we select some input point at the left-hand boundary. From there, we assign a fracture aperture width $a + \Delta a$ to the first 10 horizontal fractures in the downstream (left–right) direction. These fractures are marked in green in Fig. 5a. In order to keep the change in the net small, we chose $\Delta a \ll a_0$.

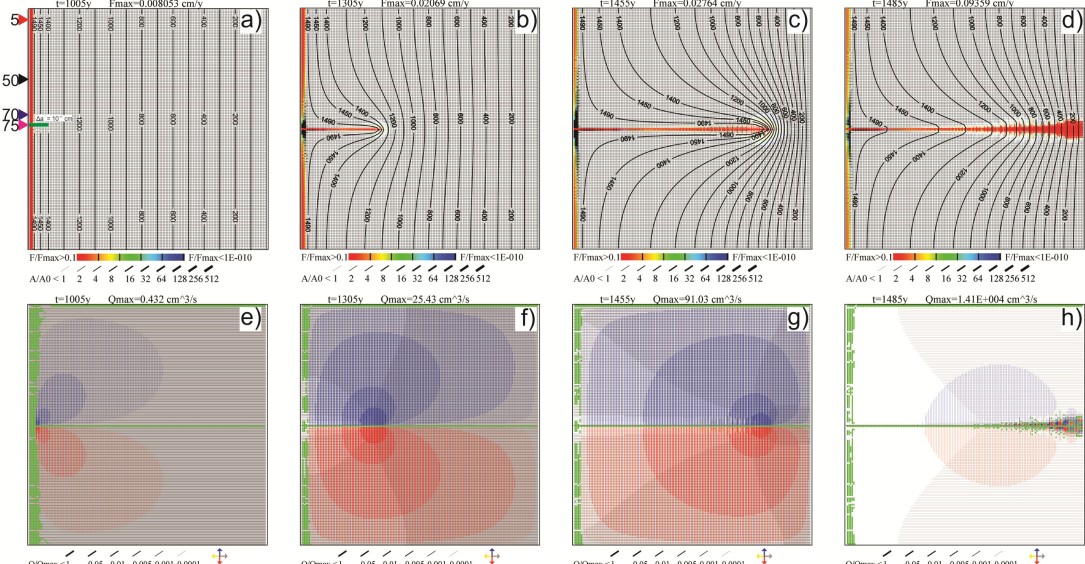

**Figure 5: Evolution of a seeded network. Same as Fig. 3, but slightly larger initial aperture by Δa = 10−11 cm are assigned to the first 10 fractures at position 75, denoted by green in a.**

Figure 5 depicts the evolution of a single wormhole initiated by a seed with $\Delta a = 10^{-9} \cdot a_0 = 10^{-11}$ cm at 75 nodes (150 m) from the upper boundary, the position along which the winning channel develops in Fig. 3. After 1005 years (Fig. 5a), the first few





horizontal fractures are widened evenly such that a sharp reaction front is visible. Most of the flow is directed left–right, but at the position of the seed a region of transverse flow is visible (Fig. 5e). From there, a wormhole invades downstream, as shown in Fig. 5b at 1305 years. Its aperture width decreases with distance from its input. Dissolution rates inside the wormhole are high, as depicted by red color but they decline rapidly beyond its tip. The direction of flow and its magnitude are depicted

by the lower panels (Fig. 5e–h). Most of the flow leaves the wormhole by flow up and down along the transverse fractures, as indicated by the thick blue and red lines close to its tip. As noted before, the line thickness gives a normalized flow rate $Q_f/Q_{max}$, where $Q_f$ is the flow through the fracture and $Q_{max}$ is the maximal flow rate occurring in some fracture in the net. This way the total flow through the colored fractures amounts to about 95% of the total flow through the domain.

With deeper penetration of the wormhole, the regions of transverse flow increase and high dissolution rates are active along

the entire wormhole (1455 years; Fig. 5c and g). At 1485 years, shortly before breakthrough, the vertical fractures in the downstream region of the wormhole have experienced widening (Fig. 5d and h). Several small competing vertical wormholes have developed in a pattern similar to the reactive front of the smooth fracture in Fig. 3. This seems to be caused by the instability inherent to the model.

The high transverse flow rates are caused by steep transverse hydraulic gradients close to the wormhole. This can be envisaged

from the lines of equal heads (isolines) shown in the upmost panels of Fig. 5. With increasing distance upstream from the tip of the wormhole, the flow rates decline because of decreasing hydraulic gradients. At the tip head, gradients in the horizontal direction rise and, consequently  the flow rates increase in this direction (grey lines). Flow rates drop along the transverse fractures with greater distance from the wormhole because at the junctions with horizontal fractures, flow is partly diverted into these horizontal fractures and then guided to the output at $h = 0$ m.

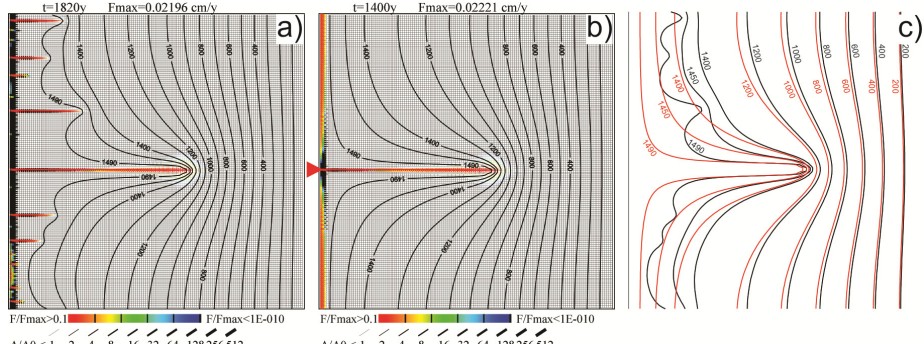

**Figure 6: Comparison for the case with no seeds (a, see Fig. 3) and single-seeded case (b, see Fig. 5) at equal length of the dominant wormholes. c) Overlaid head distribution from case with no seeds (a, black) and a seeded case (b, red).**





### 3.3 Evolution of the winning wormhole

In Fig. 6, we compare the pressure fields of the evolution in Fig. 3 (no seeds, Fig. 6a) and Fig. 5 (one seed, Fig. 6b) at times where their channels have equal lengths. In Fig. 6c, their head distributions are compared. The black isolines show the head distribution for the scenario without seeds, the red ones with one seed. In the downstream region for heads smaller than 1200

cm, the head distributions become very similar. Therefore, the evolution of the leading wormhole seems to be independent of the presence of other wormholes. On the other hand, the fingers in the non-seeded (homogenous) case (Fig. 6a), would have grown deeper into the domain if the leading channel had not existed.

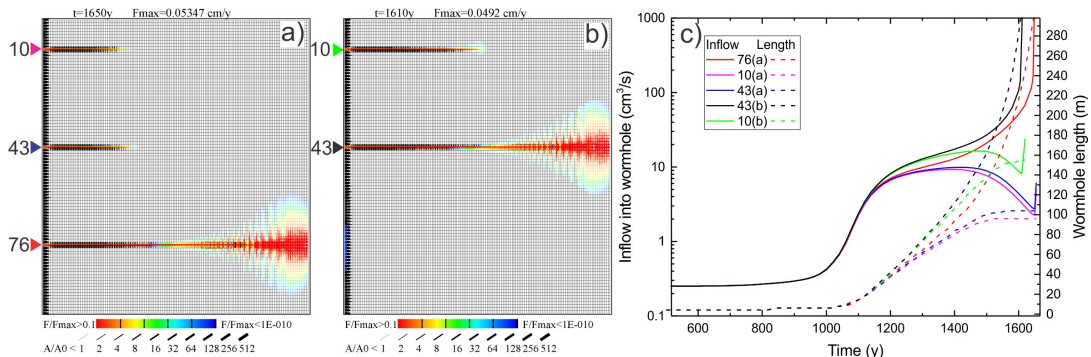

**Figure 7: Breakthrough situation in a network with (a) three seeds at positions 10, 43, and 76, and (b) two seeds at positions 10 and 43. c) Evolution of input flow rates and wormhole lengths for seeded wormholes in a and b. Note that the winning wormholes, black curve (43 b)and red curve (76 a), show a very similar behavior.**

To explore this deeper, in Fig. 7, we compare the evolution of a scenario with three seeds at $y = 10$, 43, and 76  (scenario a)

and a scenario where the previously winning seed at $y = 76$ is omitted (scenario b). The evolution in time is illustrated by the flow rates into the entrances of the wormholes (Fig. 7c).

In scenario a (Fig. 7a), until 1200 years, all flow rates are equal Fig. 7c. Then, the instability causes an advantage for the flow rate through the input at $y = 76$. The other two seeds, at 10 and 43, are inhibited and flow through them stays constant and is reduced later, whereas flow through input at $y = 76$ increases until breakthrough. If this winning seed is omitted (scenario b;

Fig. 7b), the evolution of the flow rates into the inputs of the seeds at $y = 10$ and 43 is identical to that in scenario a until 1100 years. Then, the seed at $y = 43$ gains advantage and inhibits the seed at $y = 10$. The initial evolution of the flow rates in both scenarios up to 1100 years is identical for all seeded inputs. After 1200 years, in scenario a the leading wormhole gains advantage, whereas, the competing ones stop growing. In scenario b, this event happens 80 years earlier. The evolution of the winning wormholes observed from the time of this moment is identical. This confirms the idea that their evolution is

independent of the presence of losing wormholes.





### 3.4 Evolution of a single wormhole

To understand the dynamics of the evolution of wormholes, we first investigate in detail the evolution of a single wormhole. Then, we study the competition of two wormholes, which are initiated by identical seeds at various distances from each other. Finally, a scenario with many seeds is shown.

We go back to Fig. 5, the scenario with a single seed resulting in the creation of a single wormhole. The high transverse flow rates are caused by steep transverse hydraulic gradients close to the wormhole. This can be envisaged from the head isolines shown in the upmost panels (Fig. 5a–d). With increasing distance upstream from the tip, the flow rates decline because of decreasing gradients. At the tip head, gradients in the horizontal direction rise and consequently, although small, the flow rates in this direction increase (grey lines). Flow rates drop along the transverse fractures with lateral distance from the wormhole

because at the junctions with horizontal fractures, flow is partly diverted into them and then guided to the output at $h = 0$ m.

Figure 8a illustrates the flow rates in the central fracture along the wormhole as a function of the distance from the input for various times. In the beginning, when the length of the wormhole is small, flow rates along the wormhole are low and, due to outflow into the vertical fractures, the flow rate declines to a small value at its tip, which is determined by the overall flow resistance of the initial net. With increasing time and length of the wormhole, the vertical outflow increases and, consequently,

the input flow rises. Close to the tip, due to the flow out into the transverse fractures, flow along the wormhole declines to a value determined by the remaining resistance of the net. This behaviour continues until breakthrough of the wormhole.

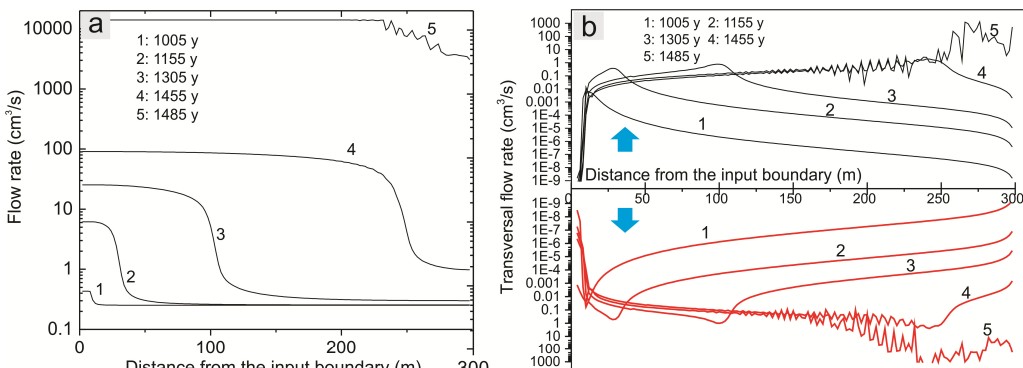

**Figure 8: a) Profiles of flow rates along a wormhole at the given times. b) Profiles of flow rates from the wormhole into the upper**
**(black) and lower (red) domain at given times.**

Figure 8b depicts the transverse flow up and down from the wormhole. In the beginning, when the wormhole is short, the transverse flow rates increase steeply by orders of magnitude until they reach a maximum close to its tip and then they decline rapidly. Note, that the sum of the total transverse outflow along the wormhole and the longitudinal outflow at its tip must be equal to the inflow at its input. The region where the horizontal flow rates decline in Fig. 8a marks the region of major




transverse outflow. When the length of the wormhole increases, this region is shifted deeper into the net and the maximum rate of flow becomes higher, marking the increase of total transverse outflow in time. Thus, the inflow of fresh solution, aggressive with respect to limestone, increases with increasing length of the wormhole supporting further dissolution along its entire length and at its tip.

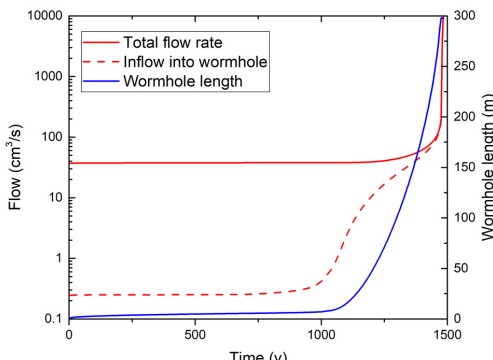

**Figure 9: Evolution of total flow through the network (full red line), input flow into the seeded wormhole (dashed) and length of the wormhole (blue line).**

Figure 9 shows the temporal evolution of the total flow through the domain, the flow rate into the input of the wormhole, and the length of the wormhole. In the beginning, flow into the wormhole is low and given by the resistance of the net. For the first

10  thousand years, flow remains almost constant. During this time, the solution front progresses evenly. Then, the instability causes initiation of the wormhole and flow through it rises. With increasing length of the wormhole, the resistance between the tip and the output becomes smaller and $Q_{tot}$ grows until at breakthrough it rises rapidly.

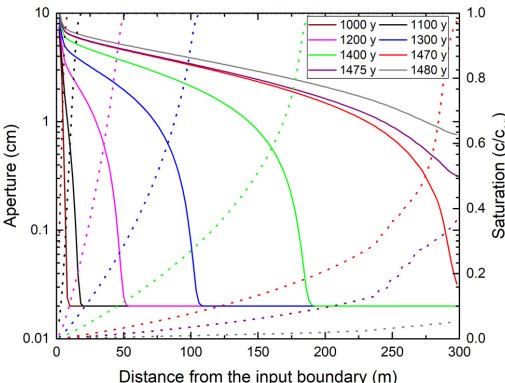

**Figure 10: Profiles of aperture widths (solid lines) and saturation ratios ($c/c_{eq}$, dashed lines), along the wormhole at given times.**





To get more insight, in Fig. 10 we have plotted the aperture widths (solid lines) and the saturation ratio $c/c_{eq}$ (dotted lines) along the line of the downstream fractures guiding the wormhole for various times of its formation. $c/c_{eq}$ increases steadily until it reaches saturation at the tip of the wormhole. There, dissolution rates drop to almost zero. The aperture widths along the first two thirds of the wormhole length are on the order of centimeters.

5   The question arises, how important is dissolution in the net adjacent to the wormhole? Is its increase of permeability sufficient to create a feedback or is it of only minor influence? To answer this question, we have investigated a scenario where only the line of horizontal fractures along the wormhole is soluble and all remaining fractures in the net are insoluble. Figure 11a depicts a comparison between the input flow rates in scenarios with and without dissolution in the net.

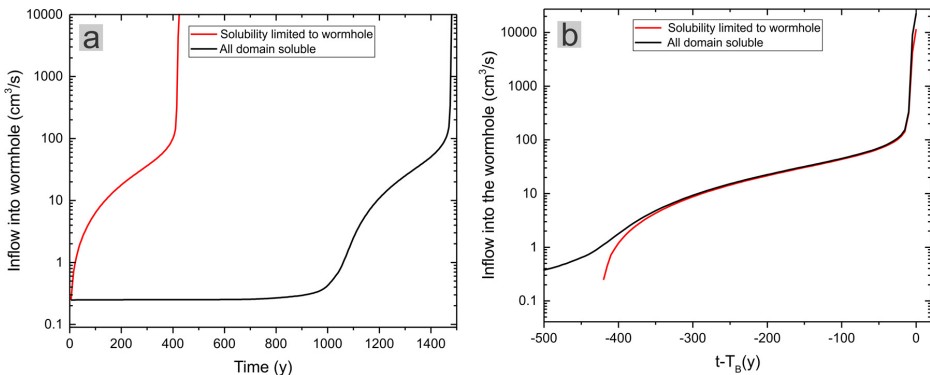

**Figure 11: a) Evolution of inflow into the wormhole for the case where solubility is limited to the position of wormhole (red line) and the case where the complete network is soluble (black line). b) Same cases as in a, but the inflow is shown as a function of time distance to the breakthrough (*t-TB*).**

For a net of soluble fractures, there is a long time of low constant flow due to the even solution front propagating slowly
15   downstream. As long as the dissolution front is completely uniform, transverse flow is not possible and enhancement of dissolution triggered by transverse outflow is absent. Each fracture behaves as an isolated fracture. As soon as the instability gives advantage to one fracture it can eject transverse flow and starts to grow rapidly until breakthrough is attained.

When the fractures of the net are insoluble, except those of the wormhole, an even dissolution front cannot be established. The line of soluble fractures along which the wormhole propagates gains advantage immediately and reaches breakthrough in a
20   much shorter time. It is important to note that the temporal evolutions of the breakthrough curves are almost identical if one compares them from the time where flow exceeds 1 cm$^3$ s$^{-1}$. Figure 11b shows an overlay of the two curves by shifting the time by $-T_b$ to $t-T_b$. This gives evidence that the resistance of the net in both cases is almost equal or, in other words, that dissolution outside the leading fracture is practically absent during the evolution of the aquifer.



We, therefore, postulate that the main mechanism causing progression of the wormhole is an increase of the input flow caused by ejection of transverse flow into the net. In conclusion, the following feedback mechanism seems to be plausible. As soon one wormhole, for whatever reasons, becomes longer than the neighbouring ones, it emits transverse flow that increases its input flow. The resulting enhanced dissolution capacity increases the length from where transversal flow can be emitted and, consequently, the amount of outflow increases (see Fig. 8) causing growing inflow. It is interesting to note that for a net of soluble fractures the advancing dissolution front retards breakthrough considerably.

### 3.5 Interaction between two wormholes

In the next step, we study the interaction of two wormholes growing simultaneously. We construct a net with two seeds, at the various positions as shown in Fig. 12, which illustrates the temporal evolution of the aquifer. We start with the two middle panels (Fig. 12b) depicting the evolution of a domain with two seeds located at $y = 60$ and $y = 90$. Until about 1000 years, all the fractures of the net widen evenly such that a sharp front of wider fractures propagates into the domain (not shown). After 1200 years, two wormholes have intruded from this front at almost equal length. In this symmetric scenario, the analytic solution would exhibit equal length of the wormholes until breakthrough. Due to the instability, however, this symmetry is broken and the wormholes develop at different pace. The transversal flow patterns up from the upper wormhole and down from it becomes strikingly different, in contrast to an isolated single wormhole where the corresponding patterns, down from the wormhole and up from it, are symmetric. Due to the presence of the second wormhole, the transversal hydraulic gradients in the region between the wormholes are reduced significantly in comparison to those in the outside regions. Because of the slightly different lengths of the two wormholes, the inflow of aggressive water into upper wormhole is lower than that into the one below. Therefore, the lower wormhole grows faster causing increased inflow.

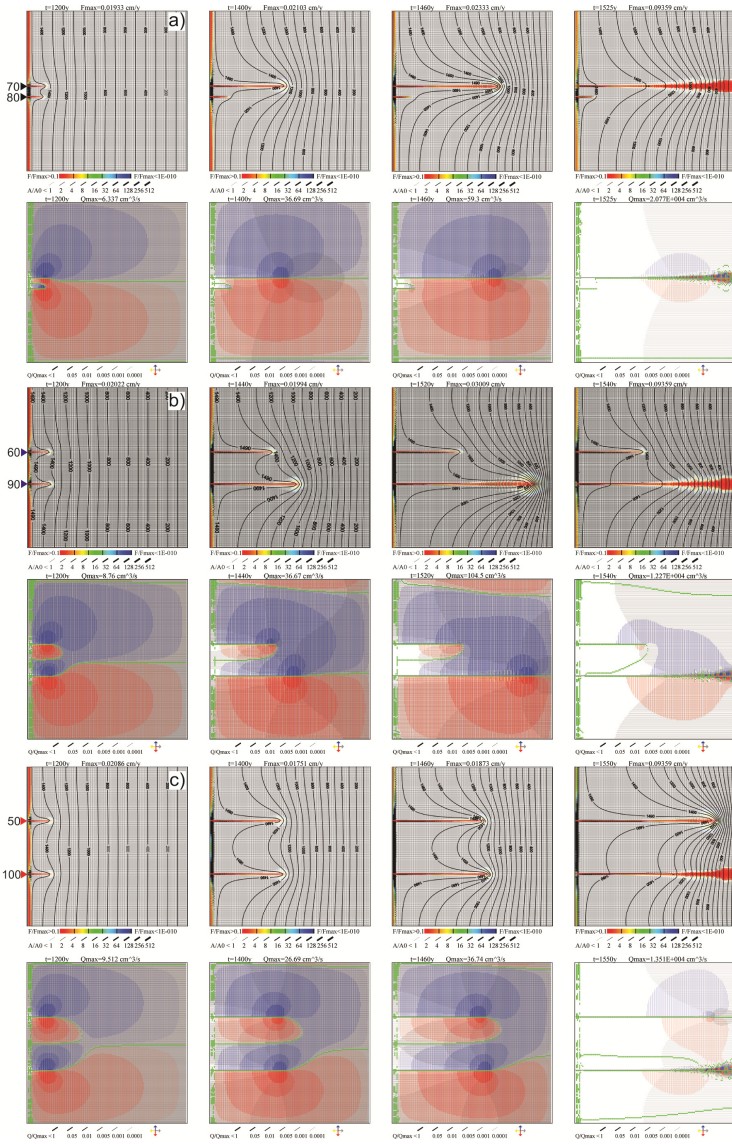

**Figure 12: Interaction of two wormholes seeded at distances of 10 (a), 30 (b), and 50 (c) nodes. First rows of panels for each case show aperture widths and relative dissolution rates, second rows show flow rates and directions. All codes and conditions as described in Fig. 3.**





Figure 13 shows the dissolution rates and flow rates along the two wormholes for various times during their evolution. As long as the length of the wormholes are close to each other (until 1200 years) their patterns are almost equal and, therefore, the flow into the two wormholes is almost equal as well. With increasing length, the inflow into the faster growing wormhole increases, whereas, that of the delayed one rises only slightly until it declines. This is reasonable because its outflow is inhibited by the

faster growing wormhole. Outflow remains low because the hydraulic gradients close to the tip of the shorter wormhole stay similar. If, however, the distance down flow between their tips exceeds some limit, this is no longer the case and the flow through the shorter wormhole decreases.

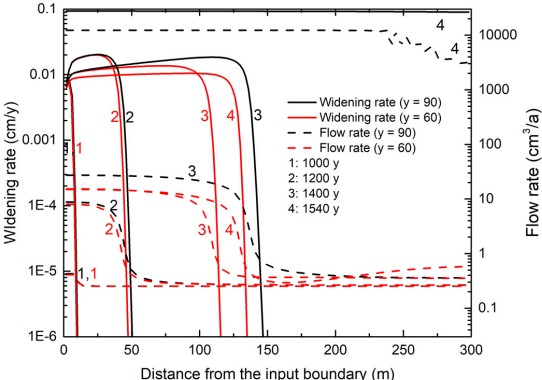

**Figure 13: Profiles of widening (full lines) and flow rates (dashed lines) for the case with seeds at $y = 60$ (red) and $y = 90$ (black) nodes**
**from the top (Fig. 12b).**

Figure 14 depicts the temporal evolution of the lengths and the input flow rates of the two wormholes for all three scenarios in Fig. 12 until breakthrough. The middle panel (Fig. 14b) depicts the scenario discussed here. In the initial state, the lengths are equal as expected. When the instability becomes active, the lower wormhole grows rapidly, whereas, the upper one experiences delayed growth. The same behaviour is exhibited by the input flow rates. This pattern is characteristic for the onset

of instability in non-linear systems.

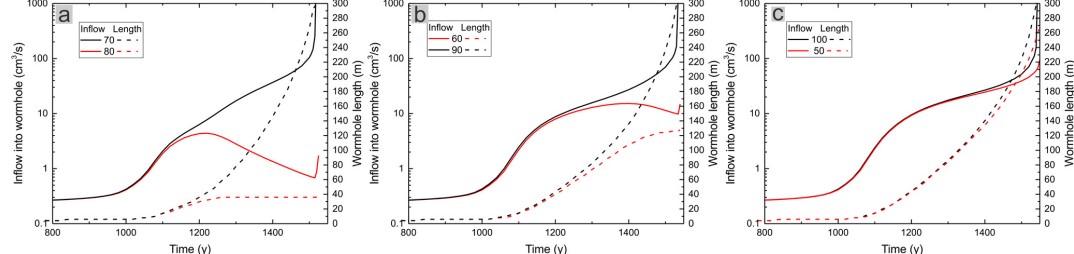

**Figure 14: Temporal evolution of flow (full lines) and length (dashed lines) of the interacting wormholes of Figure 12. Lettering corresponds to Figure 12.**



From these findings, one may conclude that the interaction between the wormholes depends on the distance between them in the *y*-direction. If this distance becomes sufficiently large that the transverse flow from both wormholes into the region between them has decayed to zero at a distance of less than half of the separation between the wormholes, we have a situation equivalent to that of an isolated wormhole. This defines a "region of influence" as suggested by (Rajaram et al., 2009). If two growing wormholes are located within this region of influence, only one of them will achieve breakthrough.

To give further evidence we study a scenario with two seeds as above, but with increased distance between them, which is larger the distance of influence. The lowest panels in Fig. 12 shows the evolution of a domain with two seeds at $y = 50$ and $y = 100$. Both wormholes invade into the net at almost equal speed until breakthrough (Fig. 12c). To illustrate the region of influence, the lower panels show flow directions by color. In addition, all fractures with flow less than $10^{-5} \cdot Q_{max}$ are shown in green. These green borders define the areas that cannot be crossed by transversal flow. They separate the domains of flow up and flow down from the wormholes. There are three borders extending from the tips downstream and one in the middle between the domains. They end in a region downstream where flow essentially becomes horizontal. With increasing length of the wormholes, this region of horizontal flow is shifted closer to the output. A further region of zero transversal flow extends from the input boundary between the wormholes. For all stages of evolution, the flow domains of both wormholes remain clearly separated and both wormholes grow independently of each other.

If, in comparison to Fig. 12b, the distance between the seeds becomes smaller as illustrated in Fig. 12a, one expects increasing dominance of the winning wormhole. At 1400 years, the two competing wormholes exhibit similar lengths in Fig. 12a and b. At later times, however, the losing wormhole stops growth in the Fig. 12a, whereas, it still gains length in Fig. 12b.

The evolution of flow and length for the three distances is shown in Fig. 14. With increasing distance the time needed to gain advantage for the winner increases until both wormholes become too distant to interact and propagate at the same pace.

### 3.6 Interaction in an array of wormholes

From these findings, we conclude that if several wormholes are located inside the region of influence of one another, only one of them will reach breakthrough. This is illustrated in Fig. 15, where 10 seeds are inserted at distances of 30 m. After 1200 years, all seeded wormholes have developed to almost equal lengths. Due to the instability, the wormholes start to grow with differing speeds. At this point, it is not possible to predict the further evolution.



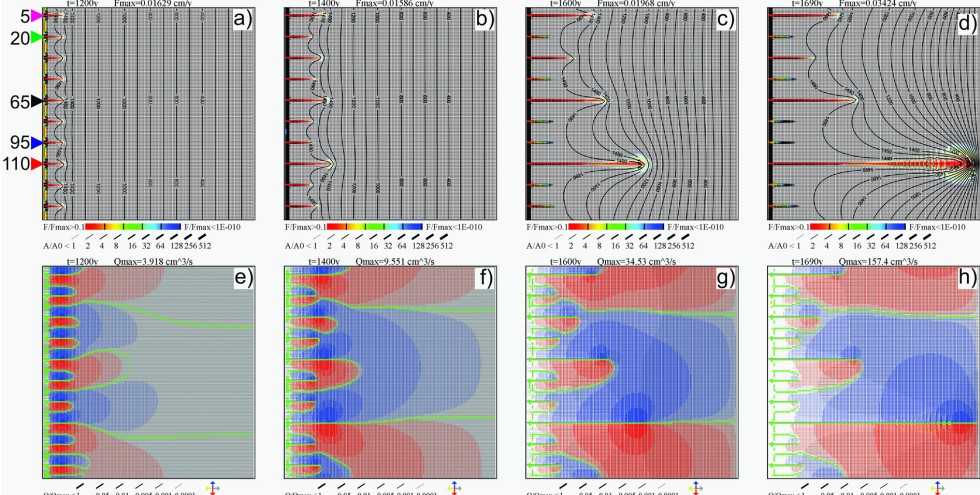

**Figure 15: Interaction of set of wormholes seeded at the distance of 15 nodes. Triangles and numbers denote location and _y_-coordinates of inputs into the selected wormholes.**

After 1400 years, three wormholes (at $y$ = 5, 65, and 110) have won a lead and have inhibited growth of all their neighbors.

5  The wormhole at $y$ = 110 has advanced further than its competitors. Therefore, these are also delayed and ultimately lose. Figure 16 shows the temporal evolution of inflow into marked wormholes and their lengths.

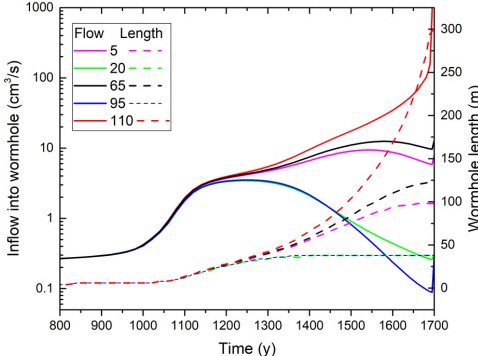

**Figure 16: Evolution of input flow rates (solid lines) and wormhole lengths (dashed lines) for the wormholes marked in Fig. 15.**

At the beginning, flow through all fractures is equal. After 1100 years, the instability causes an advantage for wormholes at $y$

10  = 5, 65, and 110 and flow rates there increase (Figs 15 and 16). Wormholes at $y$ = 20 and $y$ = 95 are in the region of influence of wormholes at $y$ = 5 and $y$ = 95 and, hence, stop growing. The same happens for wormholes at $y$ = 5 and y = 65 one hundred years later.



From what we have found so far, the following picture of the evolution of a homogeneous net as described in Fig. 3 arises. Due to the instability of the system to small perturbations (Szymczak and Ladd, 2011), the initially even propagation of all equivalent fractures breaks and some fractures gain advantage. The evolving pattern at that stage is not predictable. After some short time, however, the interaction of the different wormholes determines the head distribution and the flow rates in the entire

5  net. The further evolution proceeds in a deterministic way. Nevertheless, the final patterns are predestinated by the initial instability.

For homogeneous nets with seeds, as in Fig. 14, where an analytical solution predicts equal lengths for all wormholes starting at a seed, the final patterns show wormholes only along the directions of the seeds. From this, one can suggest that for heterogeneous nets containing percolating pathways of fractures with aperture widths differing from those in the net, the

10  wormholes should develop favorably along parts of these percolating pathways by a deterministic process where the instability plays no role.

Figure 17 gives an example of a dual network, where a net of prominent fractures with aperture widths of $A_0 = 0.03$ mm is superimposed over the homogeneous net with aperture widths of 0.02 mm. The crude net of prominent fractures is discretised into elements of 20 m by 20 m with an occupation probability of 0.72. This way percolating pathways from the input boundary to the output boundary are warranted. The prominent fractures are shown as thick lines in Fig. 17.

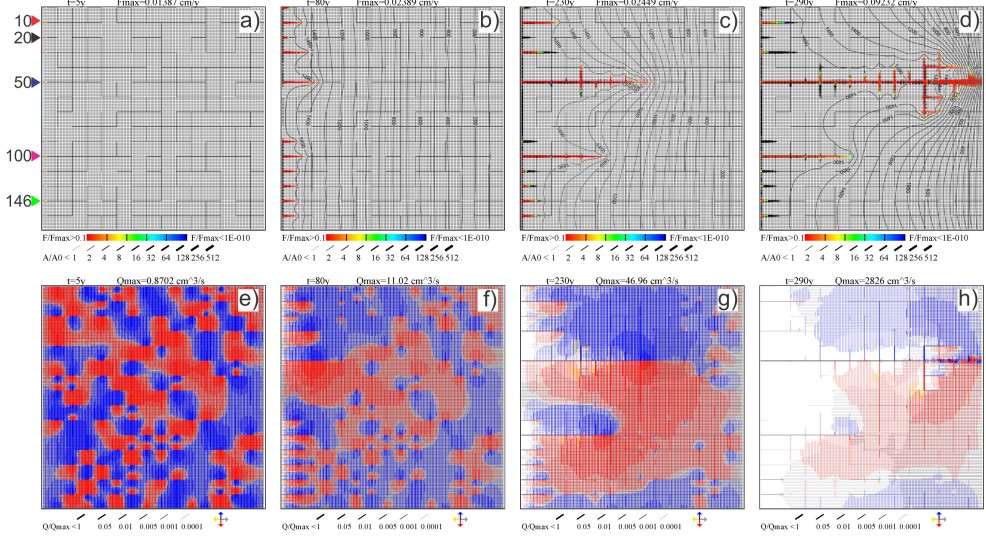

**Figure 17: Evolution of wormholes in a dual network. Network is a superposition of the basic network (Fig. 3) and a crude net of prominent fractures with $A_0 = 0.03$ cm, length 20 m and occupation probability 0.72. a–d) Aperture widths and dissolution rates. e–h) Flow rates and directions.**





After 5 years in the dual network (Fig. 17a and e), small wormholes of equal lengths have developed along all the prominent fractures originating from the input boundary because the prominent fractures act as seeds. This prevents the creation of an even dissolution front along all the inputs without prominent fractures. After 80 years (Fig. 17b and f), wormholes of differing lengths have invaded the domain. The leading one inhibits further growth of all others, as can be seen after 230 years (Fig. 17c

and g). After 290 years (Fig. 17d and h), the winning wormhole achieves breakthrough. It is interesting to note that introduction of prominent fractures reduces breakthrough time from 1900 years in the homogeneous net to 290 years (see Fig. 3). There are two reasons for this. First, an even dissolution front cannot evolve because perturbations from one-dimensionality are strong. Second, due to the wider prominent fractures along the pathway and also transverse to the winning channel, flow into its input is higher than along a pathway with fracture aperture width of 0.02 mm exclusively. Therefore, more calcite can be dissolved

and the wormhole proceeds faster to breakthrough. For completeness, Fig. 18 illustrates the flow rates into the evolving wormholes and their lengths.

Another way to break the action of the instability is to select distinct input regions or input points instead of an even head along the entire input boundary. Such cases have been explored in Dreybrodt et al. (2005a).

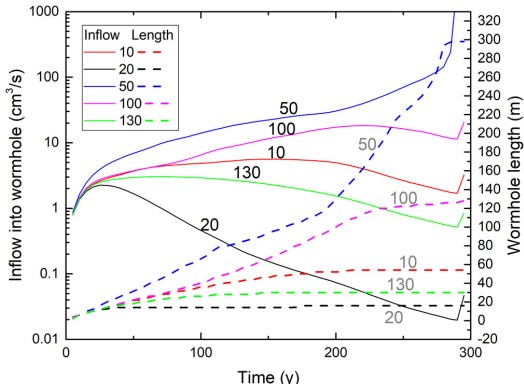

**Figure 18: Evolution of input flow rates (solid lines) and wormhole lengths (dashed lines) for the wormholes marked in Fig. 17. Numbers on curves denote the y-coordinates of the wormholes.**

## 4. Discussion

From our findings, so far we can summarize the evolution of wormholes as follows.

In Fig. 19a, we show the temporal evolution of the inflow into the winning wormhole for the basic case (see Figs. 3 and 4) in

comparison to the evolution of the same domain with one seed implanted (see Fig. 5). In both cases we find a long initial period of about 1000 years during the formation of the even dissolution front with low, almost constant inflow. In both cases the instability becomes active, but in the seeded domain this happens about 400 years earlier. After the activation of the instability, a fringe with several small channels evolves and flow rates rise slightly to a few tenths of a centimeter cubed per



second. In the unseeded case, several wormholes grow from this fringe with almost equal penetration speed until one of them has gained advantage in its length and from then on develops independently of its past history. If the net is insoluble, the evolution of a dissolution front is not possible and the wormhole starts to grow immediately. In all cases shown in Fig. 19, the evolution of the wormhole is almost identical from the moment when it has gained advantage over its competitors. This is

5 shown in Fig. 19b where we have shifted each curve by its breakthrough time $T_B$. In a nutshell, once a wormhole has gained advantage over its competitors it grows independently of its former history and its evolution is determined by the structure of the domain. This means that the evolution of the even dissolution front retards breakthrough.

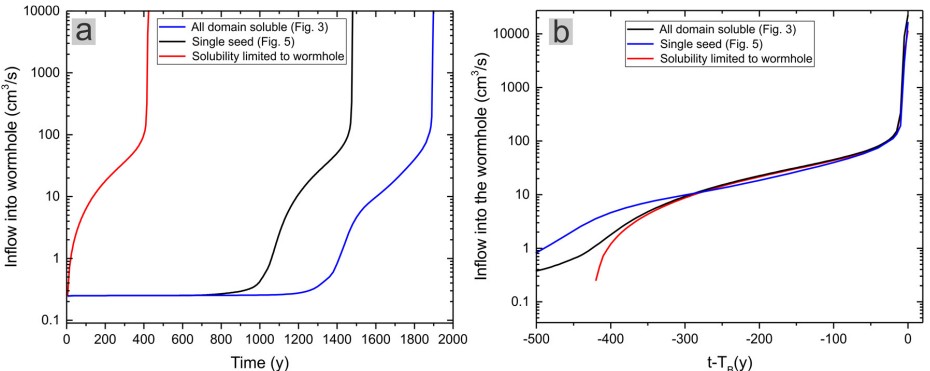

**Figure 19: Inflow into the wormhole as a function of time (a) and as a function of time distance to the breakthrough time (b) for the uniform network (blue line, Fig. 3), single-seeded case (black line, Fig. 5) and for the case where only the fractures along the wormhole are soluble (red line).**

Figure 20, as a further example, illustrates the evolution of a heterogeneous domain with statistically distributed fracture

aperture widths. These are taken from a log-normal distribution with $a_{min}$ = 0.015 cm, $a_{peak}$ = 0.02 cm, $a_{max}$ = 0.025 cm, $\sigma$ = 0.2 cm. There is no appearance of an even dissolution front but instead several competing wormholes start to grow immediately. After 300 years, one of them becomes dominant and breakthrough is achieved after only 560 years.





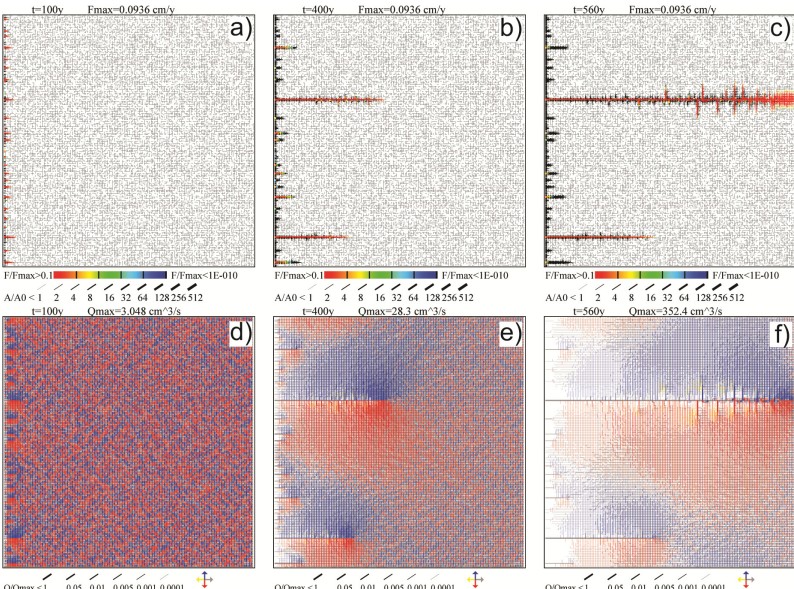

**Figure 20: Evolution of wormholes in the network with log-normal distribution of initial aperture widths. a–c) Aperture widths and dissolution rates. d–f) Flow rates and directions.**

Compared to a single 1D fracture of the same dimension (300 m x 300 m), the breakthrough times of the same fracture

5   embedded into a two dimensional array of fractures is lower by at least one order of magnitude and independent of the kinetic order of the dissolution kinetics (Dreybrodt et al., 2005b;Szymczak and Ladd, 2011).

To understand this, we go back to the evolution of the wormhole in a homogeneous aquifer with one seed. Figure 21a shows, for various times, the profiles of dissolution rates converted to widening of the fractures in centimeters per year and the concentration $c/c_{eq}$ along the fracture. In Fig. 21b the profiles of the aperture widths, the flow into the wormhole and the

10   penetration lengths are depicted. The penetration length, $\lambda$, is the distance of exponential decay from the given location, defined in Eq. 8. It is given by $\lambda = Q \cdot c_{eq} / (k \cdot P)$, $P = 2(a + b)$, where $a$ and $b$ are the aperture width and breadth, $Q$ the flow in the fracture at this position, and $c_{eq}$ the equilibrium concentration with respect to calcite. The effective reaction rate $k$ is $k = k_1 / (1 + k_1 \cdot a / (3D \cdot c_{eq}))$. Here $D$ is the constant of molecular diffusion and $k_1$ is the rate constant of the surface reaction. $(k_1 = 4 \cdot 10^{-11}$ mol cm$^{-2}$ s$^{-1}$, $c_{eq} = 10^{-6}$ mol cm$^{-3}$, $D = 10^{-5}$ cm$^2$ s$^{-1})$.





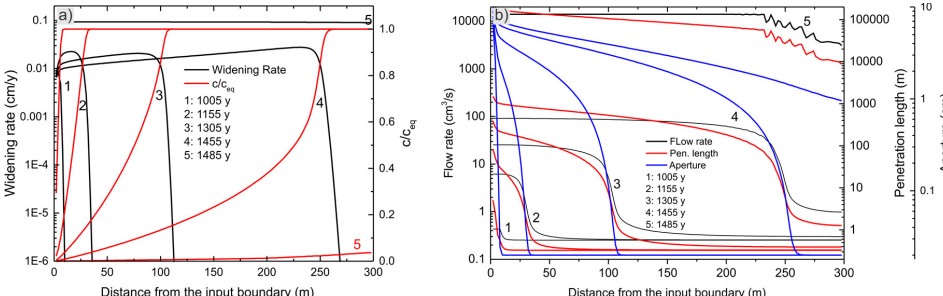

**Figure 21: Profiles of different parameters at given times for the single seeded case (network evolution presented in Fig. 5). a) widening rates (black lines, cm y$^{-1}$), saturation ratio (red lines, $c/c_{eq}$), b) flow rates (black), penetration lengths (red), and aperture widths (blue).**

With increasing $a$, $k$ decreases and the penetration lengths increase accordingly in time at all locations in the wormhole, where dissolution is active. In the upstream part, where flow is large, they are on the order of several ten to hundreds of meters. The saturation, $c/c_{eq}$, of the solution increases slowly to about 0.8 close to the tip of the evolving wormhole and widening of the fractures is active along the entire length of the wormhole. At its tip, the aperture widths decline rapidly and $k$ increases accordingly. Consequently, the penetration lengths decline to a value of a few centimeters and dissolution practically stops in

fractures beyond the wormhole.

At each location in the fracture, penetration length increases with time. Therefore, dissolution can penetrate deeper into the fracture and the length of the wormhole increases. With increasing length, the amount of outflow from the wormhole into the net increases and the flow into the wormhole grows because the effective resistance of the part downstream from its tip is reduced. Therefore, penetration lengths increase and cause deeper penetration of the wormhole into the aquifer. Here, the

feedback loop is related to the resistance of the net into which it is embedded.

We have, therefore, explored a scenario where all horizontal fractures at $y = 150$ have aperture widths $A_0 = 0.02$ cm, while the rest of the fractures have generally different initial aperture $0 < a_0 < 0.025$ cm. Figure 22 shows the dependence of the breakthrough times on the aperture widths of the net for various $A_0$. If the fractures of the network are closed ($a_0 = 0$ cm), the fractures with $A_0$ are isolated and cannot exchange any flow with the network. It is, therefore, equivalent to a single 1D fracture.

The breakthrough time is $4 \cdot 10^6$ years. With increasing $a_0$, the breakthrough time decreases to 800 years at $a_0 = 0.0175$ cm and the resistance of the net is reduced, favoring flow from the central fracture into the net. If $a_0$ comes close to $A_0$, the central fracture behaves like a seed and an even dissolution front arises. This retards breakthrough. In natural environments, the perturbations are generally large such that even dissolution fronts are prevented. Since the breakthrough times depend heavily on the unknown complex properties of the surrounding aquifer, it is not possible to predict breakthrough times. This example

shows that with increasing amount of water, which can flow from the fracture into the net, breakthrough times are reduced and





the instability does not arise because the surrounding net breaks the condition of one-dimensionality. Only when the perturbation is small, even dissolution fronts do develop. This scenario was already discussed in Dreybrodt *et al.* (2005a).

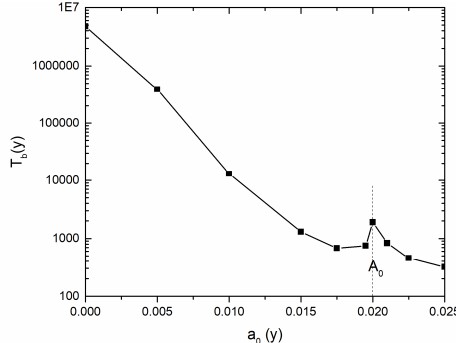

**Figure 22: Breakthrough time of a network with a central fracture $A_0$ = 0.02 cm embedded into a network of fractures with initial**
5 **aperture $a_0$ as a function of $a_0$.**

In our model, the basic element of the 2D net is treated as a 1D fracture. This has been criticised by Szymczak and Ladd(2011). They argue that due to the inherent instability, the fracture will not be widened evenly but wormholes will arise and the breakthrough time will be reduced. Wormholes, however, are created only in a limited range of Peclet and Damköhler numbers

10 (Szymczak and Ladd, 2009). The Peclet number, Pe, is defined as $Pe = a \cdot v/D$ and the Damköhler number, $Da$, as $Da = k/v$, where $v$ is the velocity of flow and is given by $v = Q/(ba)$. Another important parameter for the formation of wormholes is the penetration length of concentration, $L_p = v \cdot a/2k$. Wormholes occur for $Da > 0.01$ and $Pe$ between 10 and 1000. For $Da < 0.01$ and $Pe > 1$, however, the dissolution front is compact and uniform and the fracture can be described by the 1D approach. In our model, the initial velocity, $v_0$, is 0.125 cm s$^{-1}$ (see Figs 8a and 9). With this value one obtains $Pe = 250$ and $Da = 0.00032$

15 and the initial penetration length is $L_p = 16$ cm. With such conditions, widening of a fracture is likely in the range of a uniform compact dissolution front but wormholes cannot be excluded because the values of $Pe$ and $Da$ are close to the border between wormhole formation and uniform dissolution (Szymczak and Ladd, 2009, Fig.6). In the 2D simulation of a single wormhole (see Fig. 5), we observe the penetration of an even dissolution front limited to the first five fractures until, after 1000 years, the instability breaks this front and the wormhole starts to develop. During this time, all fractures directed downstream exhibit

20 identical dissolution and transverse flow is absent (see Fig. 9). Once the instability sets in, transverse flow rises sharply and, consequently, also the flow rates along the wormhole increase (Fig. 9). Therefore, the flow velocity in the fractures upstream from the tip of the wormhole rises by at least one order of magnitude and the penetration length takes values close to the length of the single fracture element. The $Pe$ and $Da^{-1}$ numbers rise and the evolution of the corresponding 1D fractures is shifted deeper into the region of uniform compact dissolution. In other words, all fracture elements of the net behave like 1D fractures,

25 which makes our model a reasonable approximation.



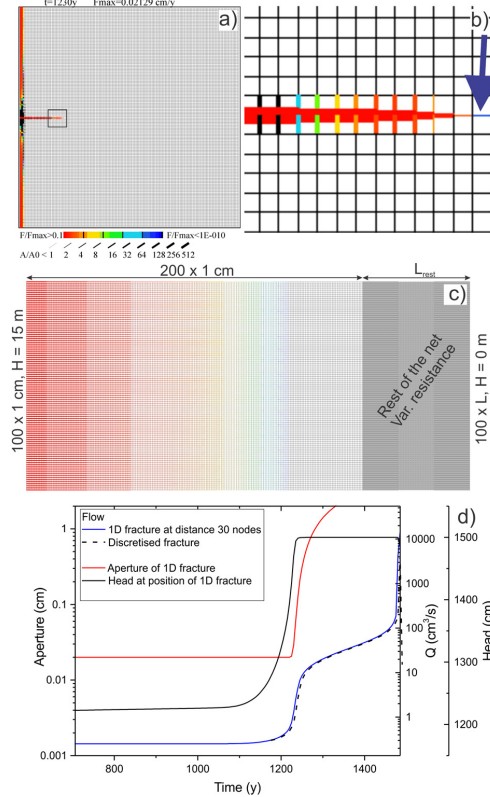

**Figure 23: a) 2D-network with penetrating wormhole. b) Excerpt of an area marked by a square in panel a. Blue arrow marks the position of a fracture discretized into 2D domain. c) 2D model of an elementary fracture. It shows a front of even dissolution. Right-hand part (grey rectangle) shows a set of parallel fractures with time dependent resistances, appended to the outlet of the 2D fracture. d) Evolution of aperture width (red line), hydraulic head at the input (black line) and flow rate (blue line) for the fracture marked by an arrow in panel b. Evolution of flow rate in a 2D analogue is shown by dashed black line.**

To verify this finding we have employed the following approach. We consider a fracture, which is just reached by the wormhole. It has experienced almost no dissolution so far. We discretize this fracture into 100 by 200 elements with dimensions of 1 cm by 1 cm and a width of 1 cm. This is illustrated in Fig. 23a that shows the 2D net with the wormhole and the even dissolution front (enlarged in Fig. 23b). The fracture of interest, at the tip of the wormhole, is marked by the blue arrow. Figure 23c depicts the dicretization of this fracture and the dissolution front that is created in it after some time. The downstream boundary nodes of the fracture are connected to insoluble fractures with length L, an aperture width of 0.02 cm, and a width of 1 cm representing the remaining downstream resistivity of the network. Figure 23c represents also the model domain for





the simulation of the discretized 1D fracture. To account for the increasing flow through this fracture, the length $L$ is chosen such that the total flow through the fracture is equal to the flow in the corresponding fracture in the 2D simulation. The simulation of the dicretized fracture is performed until the breakthrough time of the 2D model. Figure 23d depicts the evolution of hydraulic head (black line), aperture width (red line), and flow through the fracture (blue line). Until 1000 years, the head

and, consequently, the flow remain constant. Then, caused by the intruding wormhole, head and flow rise. When the wormhole has reached the fracture (at $t = 1250$ y), the head reaches its maximal value equal to the head at the input. The further increase in flow is caused by the wormhole penetrating towards the exit in the 2D simulation, reducing the resistance of the remaining net. This flow is shown by the dashed black line. In this way, we find wormholes only in the first three fractures in the 2D net located in the pathway of the wormhole. For all the other fractures, downstream along the wormhole only even and compact

dissolution fronts are observed. In other words, during the evolution of the wormhole, all fracture **elements** of the net behave like 1D fractures. Therefore, our model can be regarded as valid.

## 5. Conclusion

To reveal the mechanisms governing the evolution of wormholes in fractured limestone aquifers on the scale of several hundred meters, we have used a 2D fracture network consisting of an array of 1D fractures with defined hydrodynamic and hydro-

chemical properties.

As a basic scenario, we start with an homogeneous network where all the 1D fractures have identical initial properties. We find that the evolution of such networks proceeds in two steps. In the beginning, an even dissolution front invades slowly into the modeling domain. Dissolution in all fractures in the front is identical. Due to instability inherent in such homogeneous systems, some fractures in the front gain an advantage and dissolution penetrates deeper, thus, rendering these evolving

wormholes slightly longer than their neighbors. Wormholes develop along these advantaged fractures because, due to the differing lengths, the redistribution of hydraulic heads increases flow along these fractures and aggressive water is delivered preferentially from the input, increasing dissolution rates along these fractures. The position of the wormholes cannot be predicted deterministically. Then, the second deterministic step of wormhole formation is triggered. Several competing wormholes invade the aquifer until one of them reaches the output boundary. The other wormholes stop growing.

If one of the input fractures serves as a seed by slightly increasing its aperture width $a_0$ by only $\Delta a = 10^{-9} a_0$, only one wormhole is created. Comparison of the evolution of this wormhole with that of the homogeneous net where several wormholes start to grow, shows that the second step of wormhole evolution proceeds in a deterministic way, independently of the evolution during the first step. Inspection of the flow rates in the fractures belonging to the seeded input reveals that the trigger is switched on when flow into the lateral fractures is initiated by the instability. In the initial stage, such flow is inhibited because all fractures

are widened identically. In the second step, flow through the fractures of the evolving wormhole increases with its length because the amount of transverse flow out of the fractures depends on the remaining resistance of the net downstream the tip of the wormhole. For a 1D model of the aquifer consisting of only one 1D fracture with the same length as the 2D array, the





breakthrough time is at least one order of magnitude higher than for the 2D fracture network because lateral outflow is not possible under these boundary conditions. In summary, evolution of wormholes occurs only in 2D models as these, in contrast to 1D models, exhibit a feedback loop by utilizing the many parallel resistances in the 2D net.

Wormholes interact with each other. In a scenario with seeds at various distances, one finds a critical distance. If the separation of the wormholes is larger than this critical distance, they grow independently of each other. For smaller distances, interaction is active and the winning wormhole inhibits the growth of the losing one. If many wormholes grow initially a region of influence can be defined. If two or more growing wormholes are located within this region of influence, only one of them will achieve breakthrough.

For a heterogeneous domain with statistically distributed fracture aperture widths, taken from a log-normal distribution, there is no appearance of an even dissolution front. Instead several competing wormholes start to grow immediately. The initial step of a slowly invading even dissolution front is prevented since the homogeneity and its corresponding one-dimensionality of the net (i.e. its properties do not depend on the y-axis) is broken and transverse flow is possible from the very beginning. We find this behavior in all scenarios where the 1D properties are broken, either by inhomogeneity of the net with respect to the aperture width of the fractures or its chemical parameters (regions of insoluble fractures) or if the boundary conditions depend on the y-coordinate. In such cases, the time until breakthrough is close to the time of evolution in the second step of wormhole formation in the homogeneous scenarios. Therefore, since all natural scenarios are heterogeneous with respect to the *y*-coordinate, the evolution of an even dissolution front retarding breakthrough does not happen.

## 6. Acknowledgments

FG acknowledges the financial support from the Slovenian Research Agency (research core funding No. P6-0119). Authors thank to  Dr. Vanessa E. Johnston for a careful reading of the manuscript and pointing out many errors and inconsistencies.

## 7. Author Contributions

WD initiated the work and wrote the text. FG is the author of the model code. He performed simulations and prepared figures and figure captions. The manuscript is based on the in-depth discussions of both authors.

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
