# Peer review of "Dynamics of wormhole formation in fractured limestones"

_Hydrology and Earth System Sciences, 2018_

## Referee Comment (RC1) · A. Palmer (Referee) · 22 Jul 2018

This is a carefully prepared paper on a highly technical topic, and it sheds light on a common problem of interpretation of groundwater flow through soluble rock. None of my comments suggest errors or misunderstanding by the authors. They are only suggestions for enhancing clarity. Nearly all comments are simply to enhance the clarity of the text (English is not the primary language of either author).

PAGE / LINE 1/1 – Wormhole formation in fractured "karst aquifers" may be misleading, because it suggests mature aquifers that have already acquired solution conduits and related geomorphic features. Perhaps "fractured limestone" would be more appropriate. Karst can also develop in other lithologies, but the dynamics discussed in

this paper refer only to calcitic limestone. 2/6 – The measurements demonstrating this switch was that of Plummer and Wigley (1976) Geochimica et Cosmochimica Acta v. 40, p. 191-202. White (1977) based his suggestion on the data and interpretation in this paper. Wigley is (or was at that time) a karst scientist with specific interest in the kinetics of cave origin, although the 1976 paper did not pursue details on that topic. 2/25-26 – I agree with the authors (Dreybrodt and Gabrovsek) on this statement. 4/3 – Specifying these two concentrations assumes that calcite (limestone) is the soluble medium, as noted in line 4/17. 4/20 – Assuming standard temperature is maintained in system. 5/3 – ".. in a fracture i…." – In line 4/10, "i" is used to define the position of a node in one of the two dimensions, rather than a fracture. Meaning is clear, however. 6/18 – This is an important point, because a tiny penetration of water at zero c concentration through a very narrow opening will appear to drive the fluid to supersaturation in the model. This has given some modelers the wrong impression that no further dissolution can take place in the fissure. The authors are aware of this problem, although readers and other modelers may come to incorrect conclusions (for example, that there is a minimum aperture below which no dissolution can take place). It may be appropriate to make brief mention of this point. 7/6 – "Periodic" conditions = unclear. Would "variable" be more appropriate (i.e., varying with time)? 8/1 – "….which occurs in some fracture in the net at that time." This is unclear – does "some fracture" refer to "any fracture"? Both authors are fluent in English, but some word usages in the language are difficult even for native speakers. (See next item.) 8/4 – "transversal flow" = transverse flow (widely preferred version). Also elsewhere in paper. 9/4 - …flow into the latter… 9/6 - … also retarded and eventually stop growing. (The 'retarding' continues.) 10/22 – "Comparison between the case with no seeds (….) and….. 12/2 – "competition between two wormholes (for clarity). 12/14 – "… the vertical outflow increases and, consequently, the input flow rises." The rate of inflow is the result of greater overall efficiency of the conduit, rather than the result of increasing vertical outflow. (Both depend on continuity of flow and are the result of greater efficiency.) So a minor change in wording is suggested. Figure 8b – Are the ∼harmonic fluctuations

in the lines caused by instability in the calculated values as the result of steps in the finite-difference calculations? They may be misleading to those unfamiliar with this kind of calculation. Would a brief explanation be appropriate? 14/7 – ". . . .all remaining fractures. . . are insoluble." The meaning is clear, but of course it's the fractured medium that is insoluble (i.e., the walls of the fractures), not the fractures themselves. See also 14/18 and 15/6. Figure 11 – caption - . . . . "function of time distance to the breakthrough" is unclear. Apparently this should be "time minus breakthrough time", or (t – Tb ). 15/2 – As soon as one wormhole. . . ." 15/3 – ". . .emits transverse flow that increases its input flow." Here also, it appears that the increased inflow (and outflow) is in response to increasing overall efficiency of the conduit, rather than the result of increasing outflow at the tip. On the other hand, if water in the growing tip is being attracted by the porous medium that it is invading, as when water enters a dry sponge, then my statement is less appropriate. 15/14 – ". . . . and the wormholes develop at different rates. The transverse flow patterns, both upward and downward from the upper wormhole, are symmetric. . . . . ., the transverse hydraulic gradients. . . . 15/18 – inflow of aggressive water into the upper wormhole. . . 15/19 – add comma after "faster". 16/Fig 12 – This looks complex, but anyone reading it carefully should understand it. 17/Fig. 14 caption – Labels correspond to those in Figure 12. 17/10 - . . . .crossed by transverse flow. 18/10 – domains of upward and downward flow from the. . . . 18/13 – "horizontal" flow on graph is probably not horizontal in the real aquifer. Perhaps best to say "parallel", or "parallel to the horizontal axis". 18/18 - . . . growth in Fig. 12a 20/2 – change "even" propagation to "uniform" propagation. 20/3 – "equivalent fractures breaks down and. . . 20/8 - . . . starting at individual seeds, . . . . 20/15 - . . . .to the output boundary are expected (?) 21/8 – channel, input flow is greater than. . . (for clarity) 21/18 – From our findings so far, . . . 21/23 - . . .a few tenths of a cubic centimeter per second. 22/1 - . . almost uniform rate of penetration. . . 22/2 – If the net is insoluble. . . . – clarify to show how wormholes can develop in an insoluble net. 22/7 – for clarity, would it be better to write ". . . . development of a uniform dissolution front retards breakthrough"? 23/6 – add space: . . .2005b; Szymczak. . . . 24/16 – "horizontal" in this case means parallel to

the "X" axis. 25/16 – Meaning of uniform "compact" dissolution front = ? 26/8 – Perhaps clearer = ". . .fracture that has just been reached by the wormhole." 26/10 – Clarify "dimensions of 1 cm by 1 cm and a width of 1 cm." Should "width" refer to the larger block outlined in black, and therefore is greater than 1 cm? Or does it refer to thickness of the model? 26/14 – downstream resistivity – should this be "resistance", to distinguish it from the geophysical property of electrical resistivity? However, its meaning is clear. 27/19 - . . .dissolution along them penetrates. . . . 27/25 – "seed" fracture presumably has an initial width advantage (rather than increasing its width uniformly with time) –? Note: Because this is a simplified model used for explanation, perhaps add a comment that reference to "years" does not necessarily correspond to conditions in real aquifers.

Additional comments The value of this paper is that it shows how minor variations in initial fracture network and recharge/discharge boundaries can affect breakthrough time in an incipient karst aquifer. It shows the importance of scale in numerical modeling of such a system. The criticism of Szymczak and Ladd shows the importance of scale in these models. As we look more closely at the system, as is done in this present paper, it appears that the same concepts and functional relationships remain valid, even at a tiny scale. In conclusion, it appears that an understanding of the functional relationships of the many variables is the most valuable tool. In a real aquifer it is impossible to predict the exact initial conditions – but by recognizing the importance of heterogeneities it is possible to anticipate the controls on development of conduits within it. Or, working backward from a mature karst system, it is possible to reconstruct the initial conditions that governed the distribution of conduits. In this way, the nature of the initial aquifer and its evolution can be inferred. I am confident that the two authors have a full understanding of karst systems and of their modeling. They are the leaders in this field. However, questions posed by other authors are helpful in directing attention to potential mis-interpretations, and this exchange of ideas is healthy.

---

## Referee Comment (RC2) · P. Szymczak (Referee) · 1 Aug 2018

The manuscript "Dynamics of wormhole formation in fractured karst aquifers" looks at the evolution of apertures and permeabilities in a network of fractures. The authors show that such a network will dissolve nonuniformly, with the flow and dissolution being focused in just a few flow paths (wormholes). The authors then further analyze the growth of the wormholes, the interactions between them and the impact of heterogeneity on the pattern.

The paper is interesting, with a large number of deep insights on dissolution-driven pattern formation. Some of the model systems considered by the authors (e.g. the system with non-soluble transverse connectors or the transverse connectors of differ-

ent width than horizontal ones) are really ingenious and allow one to understand the details of the wormhole-wormhole interactions which would be hard to grasp otherwise. I have just a couple of comments (as detailed below) – the two main ones being 1) the necessity of putting these results in the context of previous work, 2) the applicability of this particular model to the real systems. To be more precise:

1. **Relevance of the previous work and references**

   Some of the scientific questions analyzed in the manuscript have been tackled before, which – in my opinion – needs to be properly acknowledged and put into context:

   (a) Upadhyay et al., J. Geophys. Res. Solid Earth, 120, 6102-6121, 2015 looks at the effect of the seeded wormhole on the dissolution pattern, compares the pattern with and without the seeded wormhole (Fig. 12 there). It also identifies the region of influence of a wormhole as a region of the width approximately equal to the wormhole length. Many of the conclusions of this paper are similar to the conclusions of the present manuscript.

   (b) There are several other systems with similar competitive dynamics, where longer fingers screen the shorter ones, thus making them grow more slowly. For example, side-branch growth in two-dimensional dendrites (see e.g Couder et al, Phys. Rev. E 71, 31602 (2005) – look at Fig. 2 there) or anisotropic viscous finger growth (Budek et al, Phys. Fluids., 27, 112109 (2015) – look at Fig. 9 there) – the authors of these papers describe the deterministic nature of growth of such systems – a property shared with the present one. A larger class of such deterministic "hierarchical growth" systems is described by the review article by J. Krug, J. Adv. in Physics, 46(2), 139-282, 1997). Additionally, in the context of wormholes, a similar system (also with deterministic dynamics) was considered by Cabeza, Y et al: Controlling factors of wormhole growth in karst aquifers, in *Hydrogeological and Environmental Investigations in Karst Systems*, pp. 379–385 (2014).

In all of these system, a hierarchical distribution of the fingers orwormholes was observed, with a number of fingers longer than $L$, scaling as approximately as $L^{-1}$. I believe this is so also in the case of the patterns analyzed in the present m/s. In fact, such a scaling is not accidental, but closely related to the observation that the region of influence of a wormhole as a region of the width approximately equal to the wormhole length – the link between the two is shown in Upadhyay et al

(c) Upadhay et al (2015) also looks at the impact of heterogeneities on the dissolution patterns in dissolving fractures (something that the authors look at in Sec. 3.6). Interestingly, their conclusions on the influence of noise on the pattern are somewhat different from those of the authors – in particular, the role of lengthscale and timescale is emphasized. As shown e.g. in Fig. 4 of this paper, whereas at the beginning the flow paths are controlled largely by heterogeneities, after a while an instability wavelength appears, on scales much larger than the characteristic correlation length of the noise (on such scales the system can again be considered uniform)

2. Applicability of the model to the real systems: the authors claim that individual fractures in the model will dissolve uniformly – I have serious doubts as to the validity of this statement. To test it, I have carried out the simulations of a single fracture element of Dreybrodt & Gabrovšek network as well as four elements of this network joined in series. I have taken the Péclet and Damköhler numbers as quoted in the m/s (btw, the estimate of the penetration length on p. 25 of the manuscript seems to be erroneous – $l_p = va/2K = a/2\mathrm{Da}$ gives $l_p = 31.25$cm for $a = 0.02$cm and $\mathrm{Da} = 0.00032$ and not $16$cm as reported in the paper). The results are presented in the attached Figures

They clearly show that even a single element of the network dissolves nonuni-

formly, contrary to what is stated on p. 25 of the m/s (where it is suggested that the dissolution in the first five elements of the network should be uniform). Nonuniform dissolution of individual elements of the network means also that one cannot impose constant pressure boundary conditions across the fracture-fracture intersections (white lines in Fig. 2), as the pressure will be highly nonuniform there, with the maximum along the developing wormhole. As a result, the pattern which would develop in such a system will consist of circular conduits and not uniformly opening fractures. The flow focusing will not change that much in this picture – our simulations indicate that the widths of the channel scale sublinearly (as $q^{1/3}$) with the flow rate, thus the conduit will still remain localized (only getting thicker with time) and the pressure will not be uniform across the fracture width. In general, I find the assumption of the uniform pressure across the lines joining fractures really dubious – it is true that such an assumption is often made in the case of network models where each link is a capillary/conduit and the intersection is supposed to be small in size compared to the capillary length; here however the width of the intersection (2m) is the same as the length of individual fractures. Imposing a uniform pressure along each such intersections (white lines in Fig. 2) changes the dynamics of the system in a dramatic way.

These remarks are not intended to diminish the value the present manuscript. As I was commenting above (point 1b), many features of flow focusing systems are rather generic and independent of the particular model. The present manuscript offers a lot of interesting insights in the dynamics of such systems and it is qualitatively correct. But I do not think that it has quantitative predictive power (in terms of breakthrough time in years etc) for the system it attempts to model (system of intersecting fractures).

**Minor comments:**

- What is the origin of factor of 3 in Eq. 7

- p. 17., line 4 – "Outflow remains low because the hydraulic gradients close to the tip of the shorter wormhole stay similar" – the meaning of this sentence is unclear

- What is plotted in Fig. 22? In the text it is stated that "...all horizontal fractures at $y = 150$ have aperture widths $A_0 = 0.02$cm, while the rest of the fractures have generally different initial aperture $0 < a_0 < 0.025$cm. Figure 22 shows the dependence of the breakthrough times on the aperture widths of the net for various $A_0$". But the caption of this Figure suggests that the results are plotted as a function of $a_0$ and not $A_0$.

**Technical comments:**

- The lettering in many figures (particularly Fig. 3, 5, 6, 7, 12, 15, 17, 20, 23) is illegible. Small bitmapped fonts are used which are almost unreadable. This is a real pity, since a lot of information is lost in this way – I urge the authors to prepare good quality figures, preferably in vector format

[Figure]

**Fig. 1.** Dissolution of a single fracture of Dreybrodt and Gabrovsek network. Colors indicate deep erosion (red), intermediate erosion (yellow and green), low erosion (blue), no erosion (black).

[Figure]

**Fig. 2.** Simulation of the dissolution of four elements of Dreybrodt and Gabrovsek network.

---

## Author Comment (AC1) · 31 Aug 2018

**Reply to the Interactive comments by A. Palmer**

We sincerely thank to the referee for taking a time for a careful reading of the manuscript and pointing to many inconsistencies and poorly formulated statements. We are very glad that the reviewer has seen the importance and potential impact of this manuscript, which gives us additional confidence in our work.

We have considered **all** comments and suggestions and did appropriate changes as suggested, which is well visible in the tracked version of the revised manuscript, where all changes are visible and those corresponding to the comments of dr. Palmer are highlighted yellow.

However, there are some comments that need a bit more discussion. These comments and responses to them are listed below:

- **2/6: The measurements demonstrating this switch was that of Plummer and Wigley (1976) Geochimica et Cosmochimica Acta v. 40, p. 191-202. White (1977) based his suggestion on the data and interpretation in this paper. Wigley is (or was at that time) a karst scientist with specific interest in the kinetics of cave origin, although the 1976 paper did not pursue details on that topic.**

    *Response: We have changed the text on P 2, L 5-10: The experiments of Wigley and Plummer (Plummer and Wigley, 1976) demonstrated a switch in the dissolution kinetics to a non-linear regime close to the equilibrium concentration of calcium ions with respect to calcite . Based on these results White (1977) suggested that such a switch–reduces the dissolution rates and causes deep penetration of dissolution power into the rock.*

    *Cited and added to the literature list: Plummer, L. N., and Wigley, T. M. L.: The dissolution of calcite in CO2-saturated solutions at 25°C and 1 atmosphere total pressure, Geochimica et cosmochimica acta, 40, 191-202, https://doi.org/10.1016/0016-7037(76)90176-9, 1976.*

    **6/18: This is an important point, because a tiny penetration of water at zero c concentration through a very narrow opening will appear to drive the fluid to supersaturation in the model. This has given some modelers the wrong impression that no further dissolution can take place in the fissure. The authors are aware of this problem, although readers and other modelers may come to incorrect conclusions (for example, that there is a minimum aperture below which no dissolution can take place). It may be appropriate to make brief mention of this point.**

    **Response**: *We have mentioned this point and cited a reference to the topic.*
    *We have added text (P 7, L 10-15) reads:*
    *Otherwise wrong conclusions can be the result as in the work of Groves and Howard (1994) who claimed that for a achieving breakthrough of a fracture a minimum aperture width is necessary.*

    *Cited and added to the literature list: Groves, C. G., and A.D. Howard, (1994) Minimum hydrochemical conditions allowing limestone cave development, Water Resour. Res., 30, 607-615*

- **7/6: "Periodic" conditions = unclear. Would "variable" be more appropriate (i.e., varying with time)?**

  *Response: As Periodic Boundary Conditions* (PBC) *are less known within the broader research community, we gave some extra explanation there. Using PBC we excluded the influence of boundaries at the top and bottom of the domain. We introduced PCB by »stitching« these two boundaries. This way the »vertical« flow entering the lower boundary continues down from the upper boundary and vice versa. As said in the text, we somehow wrap the 2D plain domain around a cylinder.*

  **At P7, L25 we have added:** *The upper and lower boundaries have have periodic conditions. Topologically this means that a 2D domain is mapped onto a cylinder. This makes the evolution of fractures independent from their distance from the upper/lower boundaries, which is not the case if these are no-flow boundaries.*

- **12/14: … the vertical outflow increases and, consequently, the input flow rises." The rate of inflow is the result of greater overall efficiency of the conduit, rather than the result of increasing vertical outflow.(Both depend on continuity of flow and are the result of greater efficiency.) So a minor change in wording is suggested.**

- **15/3: … emits transverse flow that increases its input flow." Here also, it appears that the increased inflow (and outflow) is in response to increasing overall efficiency of the conduit, rather than the result of increasing outflow at the tip. On the other hand, if water in the growing tip is being attracted by the porous medium that it is invading, as when water enters a dry sponge, then my statement is less appropriate.**

  - **Response:** *We have made minor changes in the text accordingly to make the message clearer. However, the main mechanism of the wormhole growth is the increasing transverse flow, through which a wormhole »invades« the flow field of competitors. Offering small resistance to flow, the high head from the boundary penetrates deep into the network along the wormhole, making the wormhole a high head region injecting the flow upward and downward into the adjacent fractured »matrix«. This allows high flow and dissolution rates in the wormhole. It is of course true that the resistance of the wormhole itself is decreasing and that the gradient between its tip and the outflow boundary increases as well, making it more flow efficient. However, this less effective than the transverse flow, which actually makes the difference to a 1D scenario. The referee's concept of »dry sponge« is conceptually close to what happens here, although the surrounding matric is not dry, but at lower head.*

    **At P13, L 15 the text is added:** *With increasing time and length of the wormhole, the vertical outflow increases allowing rising input flow at the constant head boundary.*

    **Also at P16, L15- the text now reads:** We, therefore, postulate that the main mechanism causing progression of the wormhole is an increase of the input flow caused by ejection of transverse flow into the net. In conclusion, the following feedback mechanism seems to be plausible. *As soon as one wormhole, for whatever reasons, becomes longer than the neighbouring ones it emits transverse flow that increases its input flow. The resulting enhanced dissolution capacity increases the length from where transverse* flow can be emitted and, consequently, the amount of outflow increases (see Fig. 8) causing growing inflow. It is interesting to note that for a net of soluble fractures the advancing dissolution front retards breakthrough considerably.

- **22/2: If the net is insoluble, clarify to show how wormholes can develop in an insoluble net.**
    - *Response: This was formulated wrongly. Of course the line of fractures with the wormhole soluble while the rest of the net is insoluble.*
    *At P23, L 0-5 the text has been changed to make the situation clear. It now reads: If only one line of fractures connecting the input to the output boundary is soluble and all other fractures in the net are insoluble competition is excluded and the evolution of a dissolution front is thus not possible, so that the wormhole starts to grow immediately.*

- **26/10: Clarify "dimensions of 1 cm by 1 cm and a width of 1 cm." Should "width" refer to the largern block outlined in black, and therefore is greater than 1 cm? Or does it refer to thickness of the model?**
    - **Response:** We have revised the text at this point to make the concept clearer. The revised paragraph on *P27, L10 now reads:*
    *To verify this finding we have employed the following approach (Fig. 23). We consider a fracture, that has just been reached by the wormhole (Fig. 23a,b). It has experienced almost no dissolution so far. We discretize this fracture into a network of 100 by 200 fractures, each 1 cm long and 1 cm wide with aperture width of 0.02 cm. (Fig. 23c). Fig. 23a shows the 2D net with the wormhole and the even dissolution front. A square marks the region enlarged in Figure 23b, where the fracture of interest, at the tip of the wormhole, is marked by the blue arrow.*

---

## Author Comment (AC2) · 31 Aug 2018

**Reply to the Interactive comments by P. Szymczak**

We thank P. Szymczak for the time invested to give some important suggestions for improvement of the paper. We highly appreciate the overall positive comments that state the importance of our research on that topic.

We have considered all comments. The replies to comments are listed bellow. he changes can also be seen in a tracked version of revised manuscript, where those corresponding to the comments of Piotr Szymczak are highlighted  green.

**1. Relevance of the previous work and references:**

We agree that there are important contributions of earlier work and we have therefore added into the text on page 3:

*Wormhole formation has been in the focus of many researchers from different fields. There are other systems with similar competitive dynamics, where fingers grow. The longer ones screen the shorter ones, thus preventing their growth.*

*Side branches of 2-D dendrites growing by diffusion limited aggregation show a similar behavior ( Couder et al, 2005)  as we find it in this paper.(Budek et al, 2015) investigated growth of anisotropic viscous fingers  in flow of immiscible fluids in a periodic, rectangular network of micro fluidic channels. Although the underlying physics is different  in both cases and from that in our work  the temporal evolution of viscous fingers is similar as we observe it in the basic case. see Fig. 3.*

*In  1997 a larger class of  systems with competitive growth is described in the review article by J. Krug, 1997) dealing with solid state properties of materials generated by molecular beam epitaxy , a topic remote from our system.*

*Most of the cited work focuses on the mathematical properties of competitive growth. Therefore they are not perceived by the community of earth science. In this work we take a different empirical approach. From the results of model realizations we detect the underlying mechanisms of hydro dynamical flow in the fractures and its interaction with dissolution widening their apertures.*

*To this end we use the idea of Upadhyay et al., 2015, who has put seeds into the entrance region of the modeling domain consisting of areas with increased fracture aperture width with respect to the apertures widths in the net. This way the seed triggers wormhole growth from its region.*

and after line 9, page 9:

*To this end we use the idea of Upadhyay et al., 2015, who has put seeds into the entrance region of the modeling domain consisting of areas with increased fracture aperture width with respect to the apertures widths in the net. This way the seed triggers wormhole growth from its region.*

We have not cited the paper of Cabeza, Y et al: Controlling factors of wormhole growth in karst aquifers, in Hydrogeological and Environmental Investigations in Karst Systems, pp. 379–385 (2014) because it deals only with the evolution of one single wormhole to characterize its capture area and does not deal with competition of wormholes.

**2. Applicability of the model to the real systems:**

We have not claimed that all fractures show even dissolution fronts. Maybe this was not stated clearly. The first five fractures show wormholes. But we regarded this as not important. However the argument that the boundary conditions do not show constant head and constant concentrations at the entry of the single fractures is important. We have used our method to find out by increasing the hydraulic head and increasing Pe if we can find a condition where **all** elementary fractures show even compact dissolution until breakthrough of the corresponding 2-D net. The result is that for hydraulic heads larger than 27.5 m we observe even dissolution fronts in **all** fractures. For this case hydraulic heads are constant at the junctions of the fractures. Then we have recalculated the basic net with a head of 27.5 m and found similar behavior in the evolution of wormholes . This means as pointed out by Prof. Szymczak that *many features of flow focusing systems are rather generic and independent of the particular model.*

We have explained this in the text, lines 3-2, pages 29-30:

*After about 1200 years in the first five 1-D fractures this even front breaks due to the instability and wormholes develop. Whereas in all the following 1-D fractures downstream, due to the increasing flow after the 2D- wormhole has arrived there the Peclet number rises sharply by about one order of magnitude within a few ten years. Therefore these 1-D fractures exhibit an even dissolution front. In our model we have assumed that in each junction of fractures lateral head differences are smoothed, such way that at the downstream input fractures constant head conditions can be applied. P. Szymczak in the interactive comment regarding this work has argued correctly that this assumption is dubious "as the pressure will be highly non uniform there, with the maximum along the developing wormhole" at the output of the fracture. Due to the computational limitations, however, we have no choice in our approximation. A better approximation might have been to limit the width of the fractures to about one tenth of the width we use to account for the wormhole formation in the first fractures. The basic behavior of the wormhole formation under these conditions will be similar qualitatively to our findings.*

*On the other hand under initial boundary conditions of higher head at the input of our 2-D model **all** fractures may show even compact dissolution fronts due to higher flow through the system. Therefore we have repeated the procedure described above for higher heads imposed onto the 2-dimensional net. For heads higher than 27.5 m we find even compact dissolution fronts in **all** fractures including the entrance one. We have also repeated the calculation for the basic case (see Fig.3) with the elevated head of 27.5 m instead of 15 m and found qualitatively similar behavior as for a head of 15 m. From this one may conclude that wormholing in the 1-D fractures does not change the general behavior of the nets because as pointed out by P. Szymczak in his interactive comment "many features of flow focusing systems are rather generic and independent of the particular model".*
We agree that our model has no quantitative predictive power (in terms of breakthrough time in years etc). We have added this to the text on page 30

*Of course our approximation cannot be applied to predict breakthrough times in real systems. The target of our work is to get insight into the processes active during the formations of wormholes.*

**3. Minor comments.**

There is a typo 3 must read 6. See Dreybrodt, 1988. We have added this citation.

Eq.7. There is a typo, 3 must read 6 close to the value as used by Szymczak   See Dreybrodt, 1988. We have added this citation.

p. 17., line 4: We have changed the text to:

*With increasing length, the inflow into the faster growing wormhole increases, whereas, that of the delayed one rises only slightly until it declines. This is reasonable because its outflow is inhibited by the faster growing wormhole. If, however, the distance down flow between their tips exceeds some limit, the flow through the shorter wormhole decreases.*

**What is plotted in Fig. 22?**

There is a typo in the text $A_0$ must read $a_0$. We have corrected this.

**Technical comments:**

We have made the small lettering because of space limitations. As the paper is electronic, the magnification allows to see details. Therefore, we prefer to leave as it stands.

---

## Author Response (AR2)

**Dear Editor,**

First, we would like to thank again for prompt consideration of this manuscript and the related material and for an immediate action. We have considered all the comments of the reviewer, which all refer to the abstract.

The following actions have been made:

- *Wormholes* are defined at the beginning of the abstract.

- A sentence has been added stating that fractures *intersect in a rectangular grid*.

- *inserting* has ben used instead of *putting* as suggested.

- in the last sentence *which* is used instead of *what.*

- We have now used "these fracture *discharge*" instead of "*eject"* (line 15) as the wormholes do not attract flow from the surrounding, but emit flow to the surrounding fracture system.

We once more thank to the reviewer for this valuable comments, which have made the abstract a bit more stand-alone.

Below, please find the new abstract with marked changed or new sentences.

Yours Sincerely,

Wolfgang Dreybrodt & Franci Gabrovšek

**Abstract.** Reactive transport in porous or fractured media often results in an evolution of highly conductive flow channels, often referred to as wormholes. The most spectacular wormholes are caves in fractured limestone terrains. Here, a model of their early evolution is presented. The modelling domain is a two-dimensional square net consisting of one-dimensional fractures intersecting each other in a rectangular grid. Fractures have given width $b$ and length $l$, to each fracture a constant aperture width, $a$, (homogeneous net) or an aperture width taken from a lognormal statistical distribution (heterogeneous net) is assigned. The boundary conditions are constant head $h$ at the input driving the water downstream to the output at $h = 0$. Linear dissolution kinetics, controlled by surface kinetics and diffusion are active. First we discuss the simple case of a homogeneous net. Two steps in its evolution are observed. In the first, all fractures are widened evenly and a homogeneous even dissolution front progresses slowly into the aquifer. The second step is triggered by an instability when, due to small perturbations, some of the foremost fractures gain length compared to the neighbouring ones. Then, these fractures discharge flow using the parallel resistances of the net. This way they attract more fresh aggressive water and their propagation is enhanced. Several wormholes (caves) are penetrating into the aquifer but only one reaches the output, whereas the others stop growing due to the redistribution of hydraulic heads

caused by the leading wormhole. The mechanisms governing the evolution of a single wormhole are explored by increasing the aperture width of one selected input fracture by $\Delta a << a$. In this case, only one single wormhole is created and inspection of the flow rates along it reveal the mechanism of flow enhancement in detail. If one uses a heterogeneous net, the first step of evolution is suppressed because of the large perturbations and wormholes start to grow immediately. We have also modelled the case of several competing wormholes in a homogeneous net by inserting appropriate seeds. We find that there is a critical distance between the wormholes. Within this distance only one wormhole survives, whereas there is no interaction between them when they are separated by more than the critical distance. We also answer the question, Why do wormholes in a two-dimensional model exhibit breakthrough times at least one order of magnitude smaller than a one-dimensional model representing the aquifer by one single plane parallel fracture of the same dimensions? Finally, we present several scenarios with non-homogeneous distribution of initial aperture widths. In these, uniform dissolution front does not develop and wormholes start to grow immediately, which is more likely expected in nature.